# Hallucination is a Consequence of Space-Optimality:
# A Rate-Distortion Theorem for Membership Testing

**Anxin Guo** [1]   **Jingwei Li** [2]

## Abstract

Large language models often hallucinate with high confidence on "random facts" that lack inferable patterns. We formalize the memorization of such facts as a membership testing problem, unifying the discrete error metrics of Bloom filters with the continuous log-loss of LLMs. By analyzing this problem in the regime where facts are sparse in the universe of plausible claims, we establish a rate-distortion theorem: the optimal memory efficiency is characterized by the minimum KL divergence between score distributions on facts and non-facts. This theoretical framework provides a distinctive explanation for hallucination under an idealized setting: even with optimal training, perfect data, and a simplified "closed world" setting, the information-theoretically optimal strategy under limited capacity is not to abstain or forget, but to assign high confidence to some non-facts, resulting in hallucination. We validate this theory empirically on both synthetic and real-world data, showing that hallucinations persist as a natural consequence of lossy compression. The same theorem recovers and sharpens classical space lower bounds for Bloom-type filters, pinning down an additive constant left open for two-sided filters.

## 1. Introduction

Despite their transformative impact, Large Language Models (LLMs) are hindered by *hallucinations*—the generation of confident and plausible yet factually incorrect statements. Recently, an influential line of work (Kalai et al., 2025; Mohsin et al., 2025; Xu, 2025; Wu et al., 2025; Gumaan, 2025) provided theoretical explanations of hallucination from a statistical learning perspective, by viewing an LLM as implementing a binary classifier over *random facts*—facts that are unstructured and cannot be logically inferred unless encountered during training, such as phone numbers or biographical details. By a no-free-lunch-style logic, since generalization is impossible on these facts, the model must make uninformed guesses on unseen random facts, leading to systematic inference errors. In the classification setting, hallucination corresponds to false positives resulting from such guesses.

As a natural workaround, researchers have promoted *abstention* (Kalai et al., 2025; Wen et al., 2025), where an LLM is expected to provide an "I don't know" response on unseen random factual queries. Theoretically, we can further simplify the task by adopting a "closed-world assumption," where all unseen potential facts are treated as non-facts at evaluation time [1]. Under this setting, an ideal LLM can be both informative and non-hallucinating on factual queries, as long as it accurately distinguishes the finitely many known facts from all other (potentially true but unknown) non-facts. This reduces hallucination mitigation to a sample-efficient memorization task.

However, empirical evidence suggests that LLMs struggle to perform this memorization task reliably. Even when permitted to abstain, models continue to generate high-confidence hallucinations and also exhibit "over-refusal" on legitimate queries (Cheng et al., 2024; Brahman et al., 2024; Zhu et al., 2025), reflecting a precision-recall trade-off. This points to a different bottleneck beyond no-free-lunch: the finite capacity of model parameters forces lossy compression of the training data, making memorization of specific, unstructured facts non-trivial.

This information-theoretic perspective connects to several recent works (Pan et al., 2025; Shi et al., 2025; Mohsin et al., 2025; Kim, 2025a;b) that attribute hallucination to distortion incurred when compressing an infinitely complex world of knowledge into a model with finite capacity. Despite these insights, two gaps remain. First, "compression causes errors" does not explain the *shape* of the errors:

[1]Computer Science Department, Northwestern University [2]Department of IEOR, Columbia University. Correspondence to: Anxin Guo <anxinbguo@gmail.com>, Jingwei Li <jl6639@columbia.edu>.

*Proceedings of the $43^{rd}$ International Conference on Machine Learning*, Seoul, South Korea. PMLR 306, 2026. Copyright 2026 by the author(s).

---

[1]Alternatively, one can assume all true facts are seen during training.

while forgetting is natural under limited capacity, it is unclear why hallucination is particularly prevalent. Second, existing compression-based explanations are either high-level informal arguments or assume an infinite number of facts, whereas our closed-world assumption restricts facts to a finite set. This motivates a more rigorous study in a simplified setting:

> *What is the theoretical explanation for high-confidence hallucinations in a closed world with finitely many random facts?*

In this work, we argue that these hallucinations are driven by the asymmetry between facts and non-facts. Specifically, if facts are sparse and randomly distributed in a vast universe of plausible statements, then an LLM with limited memory capacity will tend to accept many non-facts as facts.

To formalize this intuition, we model factuality judgment as a *membership testing* problem. We view each plausible statement as an element $i$ in a universe $\mathcal{U}$, and the set of known facts as a *key* set $\mathcal{K} \subseteq \mathcal{U}$. Given a query $i \in \mathcal{U}$, the model outputs a confidence score $\hat{x}_i \in [0, 1]$ indicating its belief that $i \in \mathcal{K}$; under this view, hallucinations correspond to non-keys being assigned high confidence. The score $\hat{x}_i$ can be evaluated under generic *error metrics* that measure the discrepancy between $\hat{x}_i$ and the true label $\mathbb{1}\{i \in \mathcal{K}\}$. The goal of a membership tester is to use a small *memory budget* to store the keys and achieve small expected error on queries. This abstraction subsumes approximate membership data structures[2] such as Bloom filters (Bloom, 1970), for which the query output is limited to $\{0, 1\}$.

## 1.1. Our contributions

**A rate-distortion theorem for membership testing.** We consider the sparse limit where $\frac{|\mathcal{K}|}{|\mathcal{U}|} \to 0$. In this regime, we identify the exact memory-error trade-off in the form of a rate-distortion theorem. Specifically, the minimum memory budget per key is characterized by the minimum KL-divergence between key and non-key output distributions.

**Theorem 1.1** (Informal, Theorems 3.1, 3.2). *Let* $\mathrm{KL}(P\|Q)$ *denote the base-2 Kullback-Leibler divergence between distributions $P$ and $Q$. To store $n$ keys in the sparse regime and reach a certain error level under generic error metrics, it is necessary and sufficient for a membership tester to store*

$$n \cdot \mathrm{KL}(\mu_K \| \mu_N) + o(n) \text{ bits of information,}$$

*where $\mu_K$ and $\mu_N$ are the distributions of scores $\hat{x}_i$ conditioned on $i \in \mathcal{K}$ and $i \notin \mathcal{K}$, respectively, that satisfy the error constraints and minimize the KL-divergence.*

---

[2]We will call these data structures "filters" for convenience.

*Moreover, score distributions of an optimal membership tester must converge to $\mu_K$ and $\mu_N$ in the sparse limit.*

KL divergence arises naturally from the asymmetry between keys and non-keys. If $X_1, \ldots, X_n \sim \mu_N$ are drawn i.i.d., then $n \cdot \mathrm{KL}(\mu_K \| \mu_N)$ roughly measures the negative log-likelihood of $X_1, \ldots, X_n$ being distributed as $\mu_K$. This value therefore quantifies how many extra bits of information are needed to force the key query output to be $\mu_K$, overcoming the default assumption that they follow the non-key distribution $\mu_N$.

**Hallucination as the optimal mode of error.** Our theory provides two perspectives on why it is *memory-efficient* for LLMs to hallucinate on random facts. First, consider a model tasked with estimating the probability $\mathbb{P}[i \in \mathcal{K}]$ that a potential fact $i \in \mathcal{U}$ is true, where the output $\hat{x}_i \in [0, 1]$ is evaluated with logarithmic or cross-entropy loss, consistent with the maximum likelihood objective and previously proposed training/evaluation methods for factual knowledge (Kadavath et al., 2022; Cheng et al., 2024).

The unique loss-minimizing query output under this metric, given a fixed memory capacity, is to assign high confidence to *all* facts, while assigning either *zero* or *fact-level confidence* to non-facts. In other words, the model simultaneously recalls all facts and hallucinates on a fraction of non-facts. We empirically verify this tendency in two complementary settings: small transformers trained from scratch on synthetic random strings (Section 4.3), and LoRA-tuned pretrained LLMs on both synthetic IDs and real-world ISBNs (Sections 4.3 and E.1). For a probability estimation task, hallucination, instead of systematic forgetting or uniform uncertainty, is the natural mode of error.

**Classifier via thresholding and two-sided filters.** Our second perspective establishes that any LLM decision mechanism based on *score thresholding*—whether the scores are derived from generative probabilities (Kalai et al., 2025) or the probability estimation output above—is subject to the fundamental memory-error trade-off of *two-sided filters*, a generalization of Bloom-type filters (Bloom, 1970) allowing both false positives and false negatives. Unlike the first perspective, this result applies to all LLM-based classification mechanisms and does not assume optimality.

Applying our result to filters, our analysis recovers and refines the various existing space lower bounds (Carter et al., 1978; Pagh & Rodler, 2001; Hurley & Waldvogel, 2007; Li et al., 2023a). We also show that a hash-based two-sided filter achieves our lower bound up to $o(n)$ bits of space overhead. As a corollary, eliminating hallucinations (false positives) on random facts is very costly without a simultaneous increase in forgetting or over-refusal (false negatives). Post-processing for factual accuracy only moves us *along* the memory-error frontier, not beyond it.

**Further discussions on the memory capacity.** Although modern models have billions of parameters, the effective memory budget for storing a particular family of random facts can be far smaller. Appendix A provides two explanations: (i) modern networks are encouraged to minimize pure memorization (through various regularization and MDL/PAC-Bayes viewpoints), and (ii) structured knowledge (e.g., language and reasoning) and random facts compete for a finite memory budget, where the *former takes precedence* during learning due to its greater impact on the training objective. Our view resonates with the "memorization is necessary" argument by Feldman et al. (Feldman, 2020; Feldman & Zhang, 2020; Brown et al., 2021): while (implicit) regularization reduces the overfitting to noise, its tendency to limit memorization can be detrimental to the model's performance on long-tail, high-entropy facts.

Crucially, this tendency is amplified by the rate-distortion frontier's shape: the marginal memory cost to eliminate the final few errors is prohibitively high, and training objectives optimizing for aggregate loss rather than perfect precision naturally tolerate the latter.

Conversely, our analysis supports the effectiveness of additional fine-tuning on unstructured random facts, which encourages the model to allocate more memory budget. Our framework also justifies why incorporating external information, such as using RAG (Lewis et al., 2020), is effective in mitigating hallucinations: memory budget is no longer a limiting factor when non-parametric memory is present.

### 1.2. Related work

Hallucination (Ji et al., 2023; Alansari & Luqman, 2026; Huang et al., 2025b) is often defined as the generation of content that is fluent and plausible but factually inaccurate, nonsensical, or unfaithful to source material. A large body of empirical works has identified causal factors throughout the entire LLM development pipeline, attributing them to *data-centric causes* (noisy web corpora) (Dodge et al., 2021; Bender et al., 2021; Perelkiewicz & Poswiata, 2024), *model-centric causes* (objectives/architectural limits of next-token prediction) (Bachmann & Nagarajan, 2025; Huang et al., 2025a; Welleck et al., 2020), and *inference-centric causes* (inference-time choices such as stochastic decoding) (Holtzman et al., 2020; Mallen et al., 2023; Lee et al., 2023; Li et al., 2023b). Our perspective is orthogonal to these studies, as we focus on the *information* perspective independent of these practical considerations.

On the theoretical side, a group of papers (Kalai & Vempala, 2024; Kalai et al., 2025; Mohsin et al., 2025; Xu, 2025; Wu et al., 2025; Gumaan, 2025) consider the *classification* perspective, as explained previously. Notably, Kalai & Vempala (2024); Kalai et al. (2025) establish that for calibrated models, the generative hallucination rate is inherently lower-bounded by the error rate of an induced classifier. This calibration assumption forces the model to assign probability mass to unseen potential facts, resulting in a regime that is fundamentally different from our closed-world setting. Meanwhile, several other works (Xu et al., 2024; Shi et al., 2025; Banerjee et al., 2024; Suzuki et al., 2025; Kalavasis et al., 2025; Charikar & Pabbaraju, 2025) consider LLMs as computable functions and discuss how incomputability gives rise to hallucinations. Some other works (Chlon et al., 2026; Pan et al., 2025; Shi et al., 2025; Mohsin et al., 2025; Kim, 2025a;b) consider LLMs as *lossy compressors* of knowledge and discuss how the distortion during compression causes hallucinations. Additionally, (Karpowicz, 2025) proves a voting-theory-style impossibility result for an LLM to eliminate hallucination while maintaining other desirable properties.

Beyond LLMs, our theory also relates to space lower bounds for static filters[3]. Carter et al. (1978) gave the first space lower bound for one-sided filters in the sparse limit, which was recently extended by Li et al. (2023a) to a more general form for nonzero $|\mathcal{K}|/|\mathcal{U}|$. Pagh & Rodler (2001) lower bounded space usage of two-sided filters which allow false negatives, but left an unspecified gap of $\Theta(1)$ per key. Hurley & Waldvogel (2007) applied rate-distortion theory to the case of fixed $|\mathcal{U}|/|\mathcal{K}|$, and gave a mutual-information style space lower bound. These bounds can be recovered as special cases of our main result.

## 2. Preliminaries

We first define membership testers and error metrics.

**Definition 2.1** (Membership tester). Given universe $\mathcal{U} = [u]$, key set size $n$, and two error metric functions, a **membership tester** $\mathcal{M}$ is a tuple of two algorithms with shared randomness:

- $\mathsf{Init}^{\mathcal{M}}$, where the tester takes input $\mathcal{K}$ and outputs $W$, a memory state (e.g. model parameters for LLMs or data structure contents for filters).
- $\mathsf{Query}^{\mathcal{M}}$, where the tester takes input $i \in \mathcal{U}$ and the memory state $W$, and returns a confidence score $\hat{x}_i \in [0, 1]$, indicating its estimate of $\mathbb{P}[i \in \mathcal{K}]$.

Let $B(\mathcal{M}) = I(W; \mathcal{K})$ be the *memory budget* (or *memory cost*) of $\mathcal{M}$, quantifying how many bits of information about $\mathcal{K}$ are stored. See Appendix A for further discussion of this quantity. From a data structure perspective, this lower-bounds the bits of space that $\mathcal{M}$ uses: $I(W; \mathcal{K}) \leq H(W)$.

*Remark* 2.2 (Permutation-invariance). To study the memory-error trade-off, it suffices to focus on *permutation-invariant*

---

[3]Static filters only store a fixed set $\mathcal{K}$. Filters that allow insertion or deletion have an additional space cost. See e.g. (Lovett & Porat, 2010; Kuszmaul & Walzer, 2024; Kuszmaul et al., 2025)

membership testers. That is, for any permutation $\pi : \mathcal{U} \to \mathcal{U}$, any key set $\mathcal{K}$ of size $n$, and any $i \in \mathcal{U}$, the distribution of query outputs is unaffected by the permutation:

$$\mathsf{Query}^{\mathcal{M}}(i, \mathsf{Init}^{\mathcal{M}}(\mathcal{K})) \stackrel{d}{=} \mathsf{Query}^{\mathcal{M}}(\pi(i), \mathsf{Init}^{\mathcal{M}}(\pi(\mathcal{K}))).$$

This assumption is standard and without loss of generality: given a membership tester $\mathcal{M}'$, we can use $\mathcal{M}'$ to define a permutation-invariant tester $\mathcal{M}$ by picking a uniformly random permutation $\sigma$ on $\mathcal{U}$ and defining:

$$\begin{cases} \mathsf{Init}^{\mathcal{M}}(\mathcal{K}) = \mathsf{Init}^{\mathcal{M}'}(\sigma(\mathcal{K})), \\ \mathsf{Query}^{\mathcal{M}}(i, W) = \mathsf{Query}^{\mathcal{M}'}(\sigma(i), W). \end{cases}$$

On a uniformly random key set $\mathcal{K}$, the expected error rates (defined below) of $\mathcal{M}$ are identical to those of $\mathcal{M}'$. Moreover, the distribution of $\mathsf{Query}^{\mathcal{M}}(i, \mathsf{Init}^{\mathcal{M}}(\mathcal{K}))$ depends only on whether $i$ is a key or non-key, not on the specific choice of $i$ or $\mathcal{K}$. Throughout the paper, we will assume all membership testers are permutation-invariant. While practical LLMs are not permutation-invariant, this assumption is a standard tool for analysis, and our lower bounds still apply.

**Definition 2.3** (Query output and error constraints). Let $d^K, d^N : [0,1] \to [0,\infty]$ be **error metrics** for keys $\mathcal{K}$ and non-keys $\mathcal{U} \setminus \mathcal{K}$. On query output $\hat{x} \in [0,1]$, the error on a single key (resp. non-key) is $d^K(\hat{x})$ (resp. $d^N(\hat{x})$).

For permutation-invariant membership tester $\mathcal{M}$, we denote its **query output distributions** on keys and non-keys by $\mu_K(\mathcal{M})$ and $\mu_N(\mathcal{M})$, respectively. We say $\mathcal{M}$ achieves error levels $\varepsilon_K, \varepsilon_N$ if it satisfies the following constraints:

$$\mathop{\mathbb{E}}_{\hat{X} \sim \mu_K(\mathcal{M})}[d^K(\hat{X})] \leq \varepsilon_K, \text{ and } \mathop{\mathbb{E}}_{\hat{X} \sim \mu_N(\mathcal{M})}[d^N(\hat{X})] \leq \varepsilon_N.$$

**Definition 2.4** ($\mu_K$ and $\mu_N$). We hereby formalize the query output distributions $\mu_K(\mathcal{M}), \mu_N(\mathcal{M}) \in \mathcal{P}([0,1])$. Let $\mathcal{K} \subseteq \mathcal{U}$ be a uniformly random subset of size $n$, and, independently of the randomness of $\mathcal{M}$, let $I \sim \mathrm{Unif}(\mathcal{K})$ and $I' \sim \mathrm{Unif}(\mathcal{U} \setminus \mathcal{K})$ be a uniformly random key and non-key, respectively. Then $\mu_K(\mathcal{M})$ and $\mu_N(\mathcal{M})$ are the laws of the corresponding query scores:

$$\mathsf{Query}^{\mathcal{M}}(I, \mathsf{Init}^{\mathcal{M}}(\mathcal{K})) \sim \mu_K(\mathcal{M})$$

$$\mathsf{Query}^{\mathcal{M}}(I', \mathsf{Init}^{\mathcal{M}}(\mathcal{K})) \sim \mu_N(\mathcal{M}),$$

where the randomness is over $\mathcal{K}$, the queried index $I$ (resp. $I'$), and the randomness of $\mathcal{M}$. By permutation-invariance (Remark 2.2), these distributions are well-defined: they depend only on whether the queried index is a key or a non-key, and not on the specific choice of $I$, $I'$, or $\mathcal{K}$.

For instance, under the false-negative/false-positive metrics $d^K(\hat{x}) = 1 - \hat{x}$ and $d^N(\hat{x}) = \hat{x}$, a standard (one-sided)

filter with false-positive rate $\varepsilon$—one that always returns 1 on keys and returns 1 on each non-key independently with probability $\varepsilon$—has score distributions

$$\mu_K(\mathcal{M}) = \delta_1 \quad \text{and} \quad \mu_N(\mathcal{M}) = (1 - \varepsilon)\, \delta_0 + \varepsilon\, \delta_1,$$

where $\delta_x$ denotes the Dirac point mass at $x$.

Intuitively, the membership tester would try to output high scores for keys and low scores for non-keys; the corresponding error metrics $d^K$ and $d^N$ would then be decreasing and increasing functions, respectively.

As examples of error metrics, for two-sided filters, $d^K$ and $d^N$ would characterize the FNR and FPR of the filter, corresponding to $d^K(\hat{x}) = 1 - \hat{x}$ and $d^N(\hat{x}) = \hat{x}$.

For factuality estimation on LLMs (Section 4.1), $d^K$ and $d^N$ characterize the log-loss on a key (fact) and a non-key (non-fact), i.e., $d^K(\hat{x}) = -\ln \hat{x}$, and $d^N(\hat{x}) = -\ln(1 - \hat{x})$.

**Assumption 2.5** (Assumptions on error metrics). Our main theorems apply to any error metrics $d^K, d^N$ that satisfy the following assumptions.

We assume that $d^K, d^N$ are nonnegative and lower semi-continuous on $[0,1]$, and perfect scores have zero error: $d^K(1) = d^N(0) = 0$. Additionally, we assume there exists some $c \in [0,1]$ achieving finite error under both metrics:

$$d^K(c) < \infty, \quad \text{and} \quad d^N(c) < \infty.$$

All logarithms in this paper are base-2 unless specified as $\ln$. We use $u$ to denote the universe size $|\mathcal{U}|$, and use $n$ to denote the key size $|\mathcal{K}|$. We use $\mathcal{P}([0,1])$ to denote all Borel probability measures on $[0,1]$.

## 3. A Rate-Distortion Theorem for Membership Testers

In this section, we state and prove our main theorems. Throughout this section, we assume a pair of fixed error metrics $d^K, d^N$ that satisfy Assumption 2.5.

For any pair of error rates $(\varepsilon_K, \varepsilon_N)$, let $\mathcal{C}_K(\varepsilon_K)$ and $\mathcal{C}_N(\varepsilon_N)$ be the feasible regions for query output distributions on keys and non-keys, respectively:

$$\mathcal{C}_K(\varepsilon_K) = \{\mu_K \in \mathcal{P}([0,1]) : \mathop{\mathbb{E}}_{\hat{X} \sim \mu_K}[d^K(\hat{X})] \leq \varepsilon_K\},$$

$$\mathcal{C}_N(\varepsilon_N) = \{\mu_N \in \mathcal{P}([0,1]) : \mathop{\mathbb{E}}_{\hat{X} \sim \mu_N}[d^N(\hat{X})] \leq \varepsilon_N\}.$$

Our first theorem characterizes the memory–error trade-off for membership testers.

**Theorem 3.1.** *Fix error metrics $d^K, d^N$ and error rates $\varepsilon_K, \varepsilon_N \geq 0$. Let $\{n_j\}, \{u_j\}$ be sequences of natural numbers such that $n_j \to \infty$ and $n_j/u_j \to 0$. For each $j$, let*

$\mathcal{M}_j$ be a membership tester for universe $[u_j]$ and key size $n_j$ that achieves error rates $(\varepsilon_{K,j}, \varepsilon_{N,j})$ under error metrics $d^K, d^N$. Suppose the error rates satisfy:

$$\limsup_{j\to\infty} \varepsilon_{K,j} \le \varepsilon_K, \quad \limsup_{j\to\infty} \varepsilon_{N,j} \le \varepsilon_N.$$

Then, the asymptotic per-key memory budget of $\mathcal{M}_j$ is at least:

$$\liminf_{j\to\infty} \frac{B(\mathcal{M}_j)}{n_j} \ge \min_{\mu_K \in \mathcal{C}_K(\varepsilon_K), \mu_N \in \mathcal{C}_N(\varepsilon_N)} \mathrm{KL}(\mu_K \| \mu_N).$$

Moreover, there exist sequences $\{u_j\}$, $\{n_j\}$, and $\{\mathcal{M}_j\}$ as described above that achieve the memory lower bound.

Our second result states that, when the KL term has unique minimizers, these minimizers define the query output distributions of optimal membership tester families.

**Theorem 3.2.** *Suppose $(\mu_K^*, \mu_N^*) \in \mathcal{C}_K(\varepsilon_K) \times \mathcal{C}_N(\varepsilon_N)$ is the unique minimizer of $\mathrm{KL}(\mu_K \| \mu_N)$. In the setting of Theorem 3.1, if $\{\mathcal{M}_j\}$ is asymptotically optimal, in the sense that*

$$\limsup_{j\to\infty} \frac{B(\mathcal{M}_j)}{n_j} = \mathrm{KL}(\mu_K^* \| \mu_N^*),$$

*then we must have $\mu_K(\mathcal{M}_j) \to \mu_K^*$ and $\mu_N(\mathcal{M}_j) \to \mu_N^*$ in Wasserstein-1 distance.*

Finally, if $\mathcal{M}$ is restricted to output values in $\mathrm{supp}(\mu_N^*)$ rather than $[0,1]$, then we can quantify how the memory lower bound converges to $\mathrm{KL}(\mu_K^* \| \mu_N^*)$ as $p \to 0$.

**Theorem 3.3.** *Let $\chi^2$ be the chi-squared divergence. In the setting of Theorem 3.2, suppose $\mu_N^*$ is supported on a finite set $\mathcal{X}$ and $\chi^2(\mu_K^* \| \mu_N^*) < \infty$. For any membership tester $\mathcal{M}$ for key size $n$ and universe $[u]$ with query outputs restricted to $\mathcal{X}$, if we fix $p = \frac{n}{u}$ and let $n, u \to \infty$, then*

$$\frac{B(\mathcal{M})}{n} \ge \mathrm{KL}(\mu_K^* \| \mu_N^*) - \frac{\chi^2(\mu_K^* \| \mu_N^*)}{2 \ln 2} \cdot p + o(p),$$

*and this bound is achievable.*

In the subsections below, we will present the main lemmas that lead to the above theorems. Most formal proofs are deferred to Section B.

### 3.1. Non-asymptotic lower bound on memory

Fix $u, n \in \mathbb{N}$. We first lower bound the necessary memory budget for achieving output distributions $\mu_K, \mu_N$ on key and non-key queries. Consider a random variable $X \sim \mathrm{Bern}(p)$, and $\hat{X}$ defined by the conditional distributions:

$$\hat{X} \mid (X = 1) \sim \mu_K, \quad \text{and} \quad \hat{X} \mid (X = 0) \sim \mu_N.$$

In the lemma below, we will use the function

$$F_p(\mu_K, \mu_N) = \frac{1}{p} I(X; \hat{X})$$

as a proxy for the per-key memory cost of a membership tester with query output distributions $\mu_K$ and $\mu_N$.

**Lemma 3.4.** *Let $\mathcal{M}$ be a membership tester for key size $n$ in universe $[u]$ with $u > n$. Let $p = \frac{n}{u}$ and let $F_p$ be as defined above. Then, the memory cost of $\mathcal{M}$ is at least:*

$$\frac{B(\mathcal{M})}{n} \ge F_p\big(\mu_K(\mathcal{M}), \mu_N(\mathcal{M})\big) - \frac{\log(8n)}{2n}.$$

*Proof.* See Section B.1. $\square$

We now state several technical properties of $F_p$.

**Lemma 3.5.** *For $p > 0$, let $F_p(\mu_K, \mu_N) = I(X; \hat{X})/p$, where $X \sim \mathrm{Bern}(p)$, $\hat{X} \mid (X = 1) \sim \mu_K$, and $\hat{X} \mid (X = 0) \sim \mu_N$. We also define $F_0(\mu_K, \mu_N) = \mathrm{KL}(\mu_K \| \mu_N)$. Then, for $p \in [0, 1)$ and $(\mu_K, \mu_N) \in \mathcal{P}([0,1])^2$:*

1. *The function $(p, \mu_K, \mu_N) \mapsto F_p(\mu_K, \mu_N)$ is jointly lower semi-continuous.*
2. *$F_p(\mu_K, \mu_N)$ is continuous in $p$.*
3. *$F_p$ is differentiable in $p$ with $\frac{\partial}{\partial p} F_p(\mu_K, \mu_N) = -\frac{\mathrm{KL}(\mu_N \| p\mu_K + (1-p)\mu_N)}{p^2}$ whenever $p > 0$.*

*Proof.* See Section B.2. $\square$

### 3.2. Memory-error tradeoff for custom error metrics

Now we define an analog of the rate-distortion function for membership testing. Fixing $d^K$ and $d^N$, for each $p \in (0, 1)$, let $R_p(\varepsilon_K, \varepsilon_N)$ be the minimum memory cost per key for given error constraints:

$$R_p(\varepsilon_K, \varepsilon_N) = \min_{\mu_K \in \mathcal{C}_K(\varepsilon_K), \mu_N \in \mathcal{C}_N(\varepsilon_N)} F_p(\mu_K, \mu_N),$$

where the minimum can be attained since, with respect to the weak-* topology, $\mathcal{C}_K(\varepsilon_K)$ and $\mathcal{C}_N(\varepsilon_N)$ are closed subsets of $\mathcal{P}([0,1])$, and $F_p$ is lower semi-continuous in $(\mu_K, \mu_N)$.

By taking minimum over all $(\mu_K, \mu_N)$ in the feasible region and applying Lemma 3.4, we immediately have the following memory lower bound for given error constraints:

**Corollary 3.6.** *Fix any pair of error metrics $d^K$ and $d^N$ and error rates $\varepsilon_K$ and $\varepsilon_N$. Suppose $\mathcal{M}$ is a membership tester satisfying the error constraints for universe $[u]$ and key sizes $n$ with $p = \frac{n}{u}$. Then, we have:*

$$\frac{B(\mathcal{M})}{n} \ge R_p(\varepsilon_K, \varepsilon_N) - \frac{\log(8n)}{2n}.$$

We show that this lower bound is achievable for fixed $p$.

**Lemma 3.7.** *Fix any $p = \frac{n}{u} \in (0, 1)$ and $\varepsilon_K, \varepsilon_N > 0$. Then, for all $\delta > 0$, there is a sufficiently large $n$ and $u$ such that there exists a membership tester $\mathcal{M}$ for universe $[u]$ and key sizes $n$ which achieves error rates $(\varepsilon_K + \delta, \varepsilon_N + \delta)$, and:*

$$\frac{B(\mathcal{M})}{n} \leq R_p(\varepsilon_K, \varepsilon_N) + \delta.$$

*Proof.* See Section B.3. □

**Proof roadmap.** Theorem 3.1 follows by combining the non-asymptotic lower bound with the achievability result above and then sending $p = n/u \to 0$. The lower bound uses compactness of $\mathcal{P}([0, 1])$ and lower semi-continuity of $F_p$ to pass to a limiting pair $(\mu_K, \mu_N)$, while achievability follows from Lemma 3.7 and continuity of $F_p$ at $p = 0$. The complete proofs of Theorems 3.1 to 3.3 appear in Sections B.4 to B.6.

## 4. Hallucination on Random Facts

Fix a universe $\mathcal{U}$ of $u$ unstructured *potential facts*; each $i \in \mathcal{U}$ is a plausible natural-language claim (e.g., detailed biographies as in Allen-Zhu & Li (2024)). The set of known facts is the key set $\mathcal{K} \subseteq \mathcal{U}$. In the *random-facts* regime that motivates our study, $\mathcal{K}$ is a subset of size $n$ drawn uniformly at random from $\mathcal{U}$. This regime intentionally isolates non-generalizable factual knowledge, forcing the LLM to behave like a membership tester for $\mathcal{K}$.

Consider an LLM that is trained (among other tasks) to distinguish facts in $\mathcal{K}$ from all other non-facts. Viewing the LLM as a membership tester for $\mathcal{K}$, Init corresponds to training on labeled data, and Query corresponds to the inference-time response (or internal evaluation) to a plausible claim $i \in \mathcal{U}$. We study two natural regimes of factuality judgment:

1. **Probability estimation.** The LLM generates a confidence score $\hat{x}_i \in [0, 1]$ as an estimate of $\mathbb{P}[i \in \mathcal{K}]$.

2. **Binary decision.** The LLM (possibly after thresholding or other post-processing) induces a binary decision $\hat{x}_i \in \{0, 1\}$ indicating whether $i$ is accepted as a fact.

In both cases, Theorem 3.1 reduces the minimum per-fact memory budget allocated to random facts, measured by $I(W; \mathcal{K})/n$, to a convex optimization problem under the corresponding error constraints. We refer to Section A for a detailed discussion of the memory budget in LLMs.

### 4.1. Probability estimation variant

In this variant, a query $i$ produces a fractional confidence $\hat{x} \in [0, 1]$. Let $\mu_K$ and $\mu_N$ denote the distribution of $\hat{x}$ when

$i$ is a fact ($i \in \mathcal{K}$) or a non-fact ($i \in \mathcal{U} \setminus \mathcal{K}$), respectively. We measure average factuality error by log-loss constraints:

$$\mathop{\mathbb{E}}_{\hat{x} \sim \mu_K} [-\ln \hat{x}] \leq \varepsilon_K, \qquad \mathop{\mathbb{E}}_{\hat{x} \sim \mu_N} [-\ln(1 - \hat{x})] \leq \varepsilon_N,$$

where a weighted sum of the two errors corresponds to binary cross-entropy loss. This evaluation metric is natural for the practical need to *quantify the confidence* $\mathbb{P}[True]$ of a factual output (Wen et al., 2025), and is consistent with proposed training/evaluation methods for studying whether LLMs "know what they know"(Kadavath et al., 2022; Cheng et al., 2024).

Our main conclusion is that, under log-loss, the *unique* optimal solution is intrinsically *asymmetric*: it drives all facts to a single high-confidence point, while forcing a nonzero fraction of non-facts to share the same point.

**Theorem 4.1.** *In the non-trivial regime where $\varepsilon_K > 0$, $\varepsilon_N > 0$, and $e^{-\varepsilon_K} + e^{-\varepsilon_N} > 1$, the unique minimizers $(\mu_K^*, \mu_N^*)$ of*

$$\min_{\mu_K, \mu_N} \mathrm{KL}(\mu_K \,\|\, \mu_N)$$

*subject to* $\mathop{\mathbb{E}}_{\hat{x} \sim \mu_K} [-\ln \hat{x}] \leq \varepsilon_K, \ \mathop{\mathbb{E}}_{\hat{x} \sim \mu_N} [-\ln(1 - \hat{x})] \leq \varepsilon_N$

*are given by*

$$\mu_K^* = \delta_{x^*}, \qquad \mu_N^* = (1 - q^*)\delta_0 + q^*\delta_{x^*},$$

*where $x^* = e^{-\varepsilon_K}$ and $q^* = \frac{\varepsilon_N}{-\ln(1 - x^*)}$.*

Figure 1 illustrates the shape of the optimal non-fact distribution in blue. The proof appears in Section C. To solve this optimization problem over probability distributions, we use variational calculus and verify the KKT conditions.

**Hallucination channel and its cost.** The optimal strategy for a given memory budget is to output a single high-confidence value $x^*$ on all facts, while assigning *the same* high-confidence value $x^*$ to a $q^*$ fraction of non-facts, forming a "hallucination channel." These non-facts are indistinguishable from facts by any downstream procedure that observes only $\hat{x}$. Moreover, the per-key memory lower bound simplifies to

$$\mathrm{KL}(\mu_K^* \| \mu_N^*) \;=\; \log \frac{1}{q^*}.$$

Hence, the hallucination probability $q^*$ is *solely determined by the memory capacity* dedicated to storing random facts, regardless of how we trade off the two types of errors.

**Resistance to thresholding.** The non-fact loss $-\ln(1 - \hat{x})$ heavily penalizes large $\hat{x}$, so one might expect $\mu_N^*$ to spread its mass over smaller or intermediate confidences, enabling hallucination removal by thresholding (perhaps at the cost of forgetting some facts). Theorem 4.1 rules this out: on the

Pareto frontier, non-facts must place an atom at the same $x^*$ used for facts. Thus, a threshold that accepts any fact must also accept these hallucinations, and any threshold that rejects any hallucination must also reject all facts.

## 4.2. Binary decision variant

We now consider *any binary decision rule* on the potential fact universe induced by the LLM. This covers thresholding the probability estimation in Section 4.1, as well as any classifier arising from generative hallucination probabilities (e.g., in Kalai et al. (2025)). Formally, any such pipeline induces a (randomized) decision $\hat{x}_i \in \{0, 1\}$ for each $i \in \mathcal{U}$, indicating either acceptance or rejection. We place error constraints on false negative rates (FNR) and false positive rates (FPR):

$$\mathbb{E}_{\hat{x} \sim \mu_K} [1 - \hat{x}] \leq \varepsilon_K, \qquad \mathbb{E}_{\hat{x} \sim \mu_N} [\hat{x}] \leq \varepsilon_N,$$

*Remark* 4.2. The FNR/FPR error constraints naturally extend to all real-valued scores $\hat{x} \in [0, 1]$. However, Theorem D.1 shows that for these error constraints, restricting attention to binary outputs $\{0, 1\}$ incurs no loss of optimality in the KL-minimization governing the memory bound.

**Theorem 4.3.** *In the non-trivial regime where* $\varepsilon_K, \varepsilon_N \geq 0$ *and* $\varepsilon_K + \varepsilon_N < 1$, *the minimizers* $(\mu_K^*, \mu_N^*)$ *of*

$$\min_{\mu_K, \mu_N} \mathrm{KL}(\mu_K \| \mu_N)$$
$$\textit{subject to} \quad \mathbb{E}_{\hat{x} \sim \mu_K} [1 - \hat{x}] \leq \varepsilon_K, \quad \mathbb{E}_{\hat{x} \sim \mu_N} [\hat{x}] \leq \varepsilon_N$$

*are given by*

$$\mu_K^* = \mathrm{Bern}(1 - \varepsilon_K), \qquad \mu_N^* = \mathrm{Bern}(\varepsilon_N).$$

We defer the full proof to Section D. Intuitively, once we reduce the KL-minimization to binary outputs, the resulting two-parameter convex program naturally has unique minimizers when the two constraints are tight.

**First-order approximation.** Since $\hat{x} \in \{0, 1\}$, we can use Theorem 3.3 and obtain a more fine-grained memory bound for small $p$. Specifically, as $p \to 0$, the difference between the memory bound and KL term is dominated by

$$\frac{p}{2 \ln 2} \cdot \chi^2(\mathrm{Bern}(1 - \varepsilon_K) \| \mathrm{Bern}(\varepsilon_N)).$$

**No non-trivial "hallucination-free" regime.** A corollary of our results is that one cannot eliminate hallucinations in a large universe without incurring a large cost. Driving hallucinations to zero corresponds to $\varepsilon_N \to 0$, but then

$$\mathrm{KL}(\mathrm{Bern}(1 - \varepsilon_K) \| \delta_0) = \infty \quad \text{for every } \varepsilon_K < 1.$$

Thus, zero FPR is only compatible with the trivial regime that rejects everything ($\varepsilon_K = 1$), or with an unbounded

memory budget. Interestingly, this also proves that there is no "reverse Bloom filter" which tolerates false negatives but not false positives. As the universe grows to infinity, it is infinitely expensive to correctly identify every non-fact.

**Thresholding only moves you along the frontier.** This memory-error frontier formalizes the trade-off between hallucination and over-refusal observed in practice (Cheng et al., 2024; Brahman et al., 2024). Theorem 4.3 is agnostic to how $\hat{x}$ is produced. Whether the decision is derived from a calibrated probability, a raw logit, a log-likelihood ratio, or a multi-stage verication chain that outputs a scalar score, the final thresholded mapping is exactly a two-sided filter with some $(\varepsilon_K, \varepsilon_N)$. Consequently, no choice of threshold can beat the memory-error frontier: a conservative approach to factual answers may reduce $\varepsilon_N$, but then either the memory budget or $\varepsilon_K$ must increase (more forgetting / abstention).

**Recovering filter space lower bounds.** By limiting output to $\{0, 1\}$, our framework recovers previous space lower bounds for filters as special cases. Setting $\varepsilon_K = 0$, our bound reduces to the classical result in Carter et al. (1978):

$$\mathrm{KL}(\delta_1 \| \mathrm{Bern}(\varepsilon_N)) = \log(1/\varepsilon_N) \text{ bits per key.}$$

Moreover, if $p$ is bounded away from 0, our rate-distortion function $R_p(\varepsilon_K, \varepsilon_N)$ corresponds to the fine-grained bound in Li et al. (2023a) for finite universes.

In the two-sided regime, we settle the gap in Pagh & Rodler (2001), quantifying $\log(1/\varepsilon_N) + \Theta(1)$ as the exact KL divergence value. Finally, our framework generalizes the results in Hurley & Waldvogel (2007) by moving from a single Hamming-distortion constraint on a Bernoulli source to separate FNR/FPR constraints on fixed composition sources.

To complete the argument for filters, Appendix D.3 gives an explicit construction of a hash-based two-sided filter that achieves the space lower bound up to $o(1)$ bits per key.

## 4.3. Experiments

We show that the predicted distributions in Section 4.1 align with output behaviors in two complementary settings: small transformers trained from scratch on synthetic random strings, and LoRA fine-tuning of a pretrained LLM on synthetic IDs and real-world ISBNs.

### 4.3.1. FROM-SCRATCH TRANSFORMERS

**Setup.** Let $\mathcal{U}$ be the set of all length-15 strings over the 26 English letters ($|\mathcal{U}| = 26^{15}$). We draw a key set $\mathcal{K} \subseteq \mathcal{U}$ by sampling $n = 15145$ strings uniformly at random without replacement. We train a 2-layer Transformer with varying parameter count (8767, 15145, and 33085) and analyze the distributions of output scores $\hat{x}_i$ when $i$ is a key/non-key.

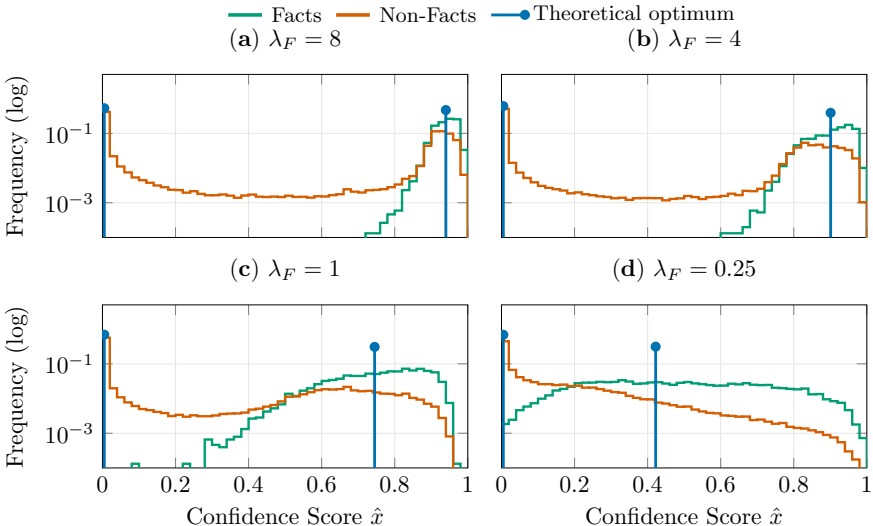

*Figure 1.* Output distributions on facts vs. non-facts ($y$-axis is in log-scale) across different choice of $\lambda_F$ for the model with 15145 parameters (1 per fact). The blue stems indicate the predicted memory-optimal atoms for non-facts under the same empirical loss values.

We train with a *weighted* binary cross-entropy loss

$$\mathcal{L} = \frac{\lambda_F}{\lambda_F + 1} \mathop{\mathbb{E}}_{i\sim\text{Unif}(\mathcal{K})}[-\ln \hat{x}(i)]$$
$$+ \frac{1}{\lambda_F + 1} \mathop{\mathbb{E}}_{i\sim\text{Unif}(\mathcal{U}\backslash\mathcal{K})}[-\ln(1 - \hat{x}(i))].$$

This loss induces the average log-losses $(\varepsilon_K, \varepsilon_N)$ on facts and non-facts, consistent with the notation in Section 4.1. Further details of model and training, as well as extra figures can be found in Appendix E. We also stress-test the sparse-regime assumption by varying the key-to-universe ratio $p$; the same qualitative picture persists, as detailed in Section E.2.

**Validation of the hallucination channel.** Figure 1 plots the empirical distributions of $\hat{x}(i)$ on facts and non-facts under different $\lambda_F$ for the model with 15145 parameters. Across all settings, the non-fact distribution is *not* only concentrated near 0; instead, it exhibits a visible high-confidence tail overlapping the fact mass, which is especially clear when the fact mass is concentrated. This qualitatively matches the hallucination channel predicted by Theorem 4.1: a non-negligible portion of non-facts must be mapped into the same high-confidence region as facts. Interestingly, a small $\lambda_F$ pushes both fact and non-fact distributions to spread out. We attribute this to the expressivity of the model architecture and properties of the sigmoid function.

**Quantitative match to the predicted atoms.** Beyond the qualitative overlap, the empirical modes in Figure 1 are comparable to the atoms $(x^\star, q^\star)$ implied by the theoretical optimizer when we plug in the observed losses $(\varepsilon_K, \varepsilon_N)$. We further compute the KL divergence between the binned

empirical distributions and find that the (discretized) learned distributions (with 50 bins) incur only an $\approx 12\%$ overhead in KL divergence, relative to the information-theoretic lower bound for the given error rates $(\varepsilon_K, \varepsilon_N)$ (Figure 2b). We also note that our estimation of effective memory is close to 2 bits per parameter, consistent with the findings in Allen-Zhu & Li (2025) when memorizing random data.

**Effect of reweighting facts vs. non-facts.** Increasing $\lambda_F$ pushes the optimizer to reduce $\varepsilon_K$ (higher recall). Consequently, the model sharpens a high-confidence region for facts, and inevitably drags a larger fraction of non-facts into the same region, causing a sharp rise in hallucination. Figure 2a reports $\varepsilon_K$, $\varepsilon_N$, and the corresponding per-key memory lower bound across the weight sweep. The heatmap shows a clear diminishing return: past a point, further improving $\varepsilon_K$ requires sacrificing $\varepsilon_N$ disproportionately, while the implied information requirement per key *decreases*—aggressive "always-recall" pressure moves the system into an especially hallucination-prone position. This suggests that it is advisable to emphasize rejecting non-facts during training – the theoretical hallucination probability in Theorem 4.1 is only a function of effective memory budget, but the budget itself depends on the training process.

### 4.3.2. LoRA FINE-TUNING OF A PRETRAINED LLM

We LoRA fine-tune (Hu et al., 2022) Qwen3.5-2B (Team, 2026) on $|\mathcal{K}| = 10000$ facts from two contrasting domains: structure-free 13-digit decimal IDs ($|\mathcal{U}| = 10^{13}$) and real-world ISBN-13 strings reservoir-sampled from the Open Library editions dump. ISBN non-facts are constructed to match real publisher-prefix statistics while remaining

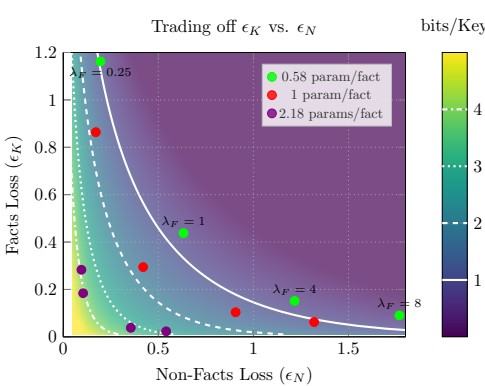

*(a)* Memory-error frontier vs. different weights.

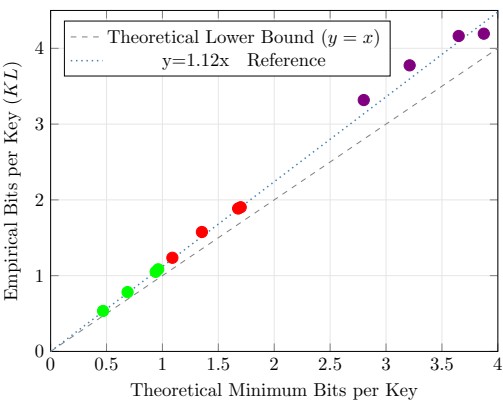

*(b)* KL divergence between empirical distributions vs. minimum information for given loss.

*Figure 2.* Effect of different weight $\lambda_F$ on fact. Each color represents a different model size. The amount of information per key decreases with $\lambda_F$ for each model size.

disjoint from $\mathcal{K}$ (full setup in Section E.1). We sweep $\lambda_F \in \{0.5, 1, 2, 4\}$ across three configurations—Synthetic at LoRA rank 4, ISBN at rank 4, and ISBN at rank 2—yielding twelve runs.

**Frontier persists with pretraining prior.** Figure 3 reports the empirical KL for the twelve runs and shows that they cluster along a single line of slope $\approx 1.18$, attaining KL within 14–22% of the lower bound implied by Theorem 4.1 at their observed $(\varepsilon_K, \varepsilon_N)$. Notably, the prior-bearing ISBN runs and the prior-free Synthetic-ID runs land on the *same* overhead band. The main visible effect of pretraining is instead optimization: compared with training from scratch, the fine-tuned model starts much closer to a useful solution and enters the stable-loss regime much faster, but this speedup does not measurably shift the final rate–distortion frontier. This is consistent with Theorem 3.1: the memory cost is governed by the information content of $\mathcal{K}$ relative to $\mathcal{U}$, not by surface regularities a pretrained model already

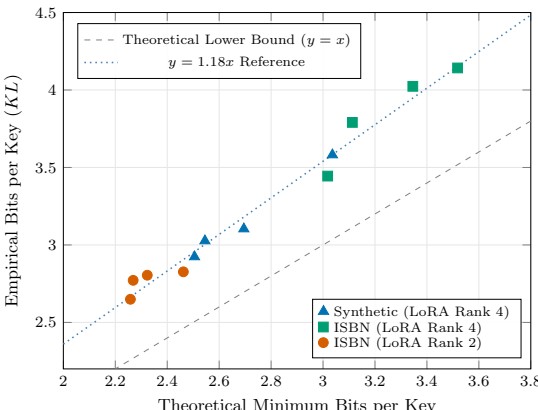

*Figure 3.* Empirical KL divergence vs. information-theoretic minimum bits per key for twelve LoRA fine-tuning runs of Qwen3.5-2B (Synthetic-IDs at rank 4; ISBN at ranks 4 and 2; $\lambda_F \in \{0.5, 1, 2, 4\}$). The runs cluster along the dotted reference line $y = 1.18\,x$, indicating an average $\approx 18\%$ overhead relative to the lower bound from Theorem 4.1. Representative confidence distributions and training dynamics appear in Section E.1.

encodes.

**Effective memory is much smaller than trainable parameter count.** In absolute terms, the empirical KL stored per trainable LoRA parameter is $\approx 0.17$ bits/param at rank 4 and $\approx 0.27$ bits/param at rank 2. Both are an order of magnitude below the $\approx 2$ bits/param attained by our from-scratch models (Allen-Zhu & Li, 2025). This is unsurprising—LoRA can only express a low-rank update on top of frozen base weights, so it mostly makes local adjustments around the pretrained predictor rather than globally reshaping the model to fit a new random fact table—but it is a direct empirical illustration of the distinction made in Section A: the memory lower bound in Theorem 3.1 is the information capacity the model actually allocates to $\mathcal{K}$, not its nominal trainable parameter count. Full setup details and representative diagnostics appear in Section E.1.

## Acknowledgments

We thank Aravindan Vijayaraghavan, Konstantin Makarychev, and Clifford Stein for helpful discussions. Anxin Guo is partially supported by the National Science Foundation through the NSF grants ECCS-2216970 and CCF-2154100 through Aravindan Vijayaraghavan. Jingwei Li is partially supported by NSF grant CCF-2218677 and ONR grant ONR-13533312 through Clifford Stein.

## Impact Statement

This paper presents work whose goal is to advance the field of machine learning. There are many potential societal consequences of our work, none of which we feel must be

specifically highlighted here.

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

# A. Detailed Discussion of the Information Budget for Random Facts

We provide a self-contained argument in support of our modeling choice: namely, that the effective memory budget available for storing random facts is orders of magnitude smaller than the raw parameter count.

## A.1. Setup: random facts, training data, and learned weights

We model LLM training via a Markov chain relating several random variables. Let $\mathcal{K}$ denote a set of random facts (e.g., a specific list of biographies) drawn from a large universe of potential facts. Let the tuple $(\Theta, \mathcal{K})$ denote a "world state" that parameterizes the training data generation process. Let $Z$ denote the training data, and let $W$ denote the learned weights obtained by applying a randomized training algorithm to $Z$. This induces the Markov chain:

$$(\Theta, \mathcal{K}) \to Z \to W. \tag{1}$$

Our goal is to upper-bound $I(W; \mathcal{K})$, the mutual information between the learned weights $W$ and the random facts $\mathcal{K}$. This mutual information, which indicates how much the model memorizes $\mathcal{K}$, corresponds to the "bits of memory" discussed in the main body.

We present two complementary arguments. First, regularization techniques and learning-theoretic pressures limit pure memorization. Second, much of the model's effective capacity is "crowded out" by structured knowledge, leaving little budget for unstructured random facts.

## A.2. Regularization limits memorization

By the data processing inequality, we have $I(W; \mathcal{K}, \Theta) \leq I(W; Z)$; moreover, standard compression- and generalization-based perspectives suggest that $I(W; Z)$ can be far smaller than the raw parameter count.

Some of the most established theories of generalization in deep learning include the minimum description length (MDL) principle (Rissanen, 1978; Gr"unwald, 2007; Gr"unwald & Roos, 2019), PAC-Bayes (McAllester, 1999; 2003; Catoni, 2007; Alquier, 2021), and mutual-information generalization bounds (Xu & Raginsky, 2017; Bu et al., 2020; Steinke & Zakynthinou, 2020). These ideas are interconnected (Blum & Langford, 2003; Achille & Soatto, 2018; Hellstr"om et al., 2025; Grunwald et al., 2021), and they all roughly establish that *limited memorization of training data* is a key factor in generalization.

Their bounds all relate to the quantity $I(W; Z)$, the mutual information between the weights $W$ and the training data $Z$.

- Mutual-information bounds directly connect the expected generalization error to variants of $I(W; Z)$.

- PAC-Bayes bounds advocate controlling $\mathrm{KL}(Q_Z \| P)$, where $P$ is some data-independent prior over $W$ such as a standard Gaussian, and $Q_Z$ is the distribution of $W$ after training on data $Z$. This relates to mutual information via $I(W; Z) \leq \mathbb{E}_Z[\mathrm{KL}(Q_Z \| P)]$, where equality is obtained iff $P$ is the marginal distribution of $W$ induced by the learning algorithm and $Z$.

- MDL principle is based on minimizing the description length of $W$ (while keeping the training error small), which is often equivalent to minimizing $\mathrm{KL}(Q_Z \| P)$, since the latter is the extra bits needed to describe $W$ beyond the prior $P$.

These theories posit that generalization is closely tied to controlling memorization, as quantified by $I(W; Z)$. In practice, $I(W; Z)$ is shaped by the implicit bias of common regularization mechanisms (e.g., mini-batch SGD, weight decay, dropout, and early stopping) and by the preference for flat minima. If this line of reasoning is correct, then we expect training to reduce $I(W; Z)$ whenever doing so does not significantly increase the training loss.

The above theories have also given rise to a series of non-vacuous generalization bounds for deep networks (Dziugaite & Roy, 2017; 2018; Pérez-Ortiz et al., 2021) and language models (Lotfi et al., 2024). These empirical bounds are derived via quantizing and compressing the model parameters $W$, and then applying MDL/PAC-Bayes bounds. If these compressed networks maintain the same performance on the training data $Z$, then these studies provide a strong argument that $I(W; Z)$ is indeed small.

It is also worth noting that a line of work (Feldman, 2020; Feldman & Zhang, 2020; Brown et al., 2021; Feldman et al., 2025) argues that memorization of samples (and even noise) is *necessary* for generalization in certain cases. Our framework

complements this perspective: limiting memorization via (implicit or explicit) regularization, while intended to filter out noise, also inevitably excludes incompressible long-tail random facts. Under an insufficient memory budget, our theory predicts that the model is forced to hallucinate as the information-theoretically optimal response.

## A.3. Structured vs. random facts: a formal tradeoff

We now formalize the intuition that "structured knowledge" and "random facts" compete for the same limited information budget in $W$, and that structured knowledge often gets prioritized.

**Decomposition.** Suppose that the world is parameterized by random variables $(\Theta, \mathcal{K})$, where $\Theta$ represents structured components (e.g., linguistic syntax, logical rules, universal regularities) and $\mathcal{K}$ represents random facts. We assume that $\mathcal{K}$ is independent of $\Theta$ (so $I(\mathcal{K}; \Theta) = 0$).

This decomposition aligns with the **Syntax–Knowledge generative framework** recently proposed by (Pan et al., 2025), who model language generation as a hierarchical process combining a parametric *syntax model* with a non-parametric, long-tail *knowledge model*. Similar distinctions have also been explored in prior literature to explain learning dynamics, such as the disentanglement of syntax and semantics (Gong et al., 2025) and the separation of grammatical structure from content in generative models (Dyer et al., 2016; Kusner et al., 2017).

**Budget Splitting Argument.** Information theory dictates that these two components compete for the model's finite capacity. If we assume $\mathcal{K}$ is independent of $\Theta$, then a standard decomposition of multivariate information (Williams & Beer, 2010) gives:

$$
\begin{aligned}
I(W; \mathcal{K}, \Theta) &= I(W; \Theta) + I(W; \mathcal{K} \mid \Theta) \\
&= I(W; \Theta) + \underbrace{H(\mathcal{K} \mid \Theta)}_{=H(\mathcal{K})} - H(\mathcal{K} \mid W, \Theta) \\
&= I(W; \Theta) + H(\mathcal{K}) - H(\mathcal{K} \mid W) + H(\mathcal{K} \mid W) - H(\mathcal{K} \mid W, \Theta) \\
&= I(W; \Theta) + I(W; \mathcal{K}) + \underbrace{I(\mathcal{K}; \Theta \mid W)}_{\geq 0}.
\end{aligned}
$$

If the total effective capacity $I(\Theta, \mathcal{K}; W)$ is bounded, this equality implies a tradeoff:

$$
\underbrace{I(\mathcal{K}; W)}_{\text{Capacity for random facts}} \leq \underbrace{I(\Theta, \mathcal{K}; W)}_{\text{Total Budget}} - \underbrace{I(\Theta; W)}_{\text{Capacity used for structure}} .
$$

Consequently, any mechanism that increases $I(\Theta; W)$—i.e., learning more structure—necessarily reduces the remaining budget available for encoding the random residual $\mathcal{K}$.

**Prioritized Learning Dynamics.** Crucially, this is not just a static tradeoff but a dynamic one where $\Theta$ takes precedence. (Pan et al., 2025) demonstrate theoretically (via Kolmogorov Structure Functions) and empirically that LLMs exhibit frequency-dependent learning: they prioritize compressing pervasive syntactic structures ($\Theta$) before acquiring rarer factual knowledge ($\mathcal{K}$). This observation is consistent with the two-stage training dynamics of Transformers (Gong et al., 2025), as well as the empirical difficulty for trained models to recall high-entropy facts compared to structured knowledge (Huang et al., 2025c). Arpit et al. (2017) also concluded that regularized SGD for deep networks would prioritize learning low-entropy "simpler" samples over high-entropy noise.

Because $\Theta$ corresponds to high-frequency patterns that offer the greatest training loss reduction, real-world models "spend" their information budget on $\Theta$ first. Random facts $\mathcal{K}$, which often appear as incompressible noise to a capacity-constrained model (Pan et al., 2025), are relegated to the residual information budget. This justifies our focus on bounding $I(\mathcal{K}; W)$: it is the marginal capacity left for the long tail of world knowledge after the model has learned the essential structures of the world.

Pan et al. (2025), whose syntax–knowledge framework we drew on above, measure this budget directly. Generating 400,000 synthetic profiles with 50 templates per attribute type, they find a sharp frequency threshold: their largest model ($\sim$253M parameters) reaches only $\sim$40% accuracy on entities that appear just four times in training, while a 7.2M model hallucinates

almost all of them. Below this threshold the model does not fall silent; it fabricates, emitting grammatically clean but factually wrong profiles. That is exactly the capacity-limited error mode our framework predicts, and their mutual-information-based predictions match these measurements closely. The same pattern underwrites our regime-relevance claim: for the long-tailed, random-looking facts on which hallucination is most common, such as specific legal cases or bibliographic entries, the effective budget $I(W; K)$ is indeed very small. Huang et al. (2025c) and Arpit et al. (2017), cited above, reach the same conclusion through memorization measures not built on mutual information, finding that rare and random examples are learned last.

# B. Proofs for Section 3

## B.1. Proof of Lemma 3.4

For completeness, we first prove the lemma below.

**Lemma B.1.** *Let $X$ be a discrete random variable with $H(X) < \infty$ and let $Y$ be any random variable. Then, the conditional entropy $H(X \mid Y)$ is concave in the joint distribution of $(X, Y)$.*

*Proof.* Let $P_0$ and $P_1$ be two joint distributions on $\mathcal{X} \times \mathcal{Y}$, and let $P_\lambda = \lambda P_1 + (1 - \lambda)P_0$ for $\lambda \in [0, 1]$. We choose a reference measure $\nu$ on $\mathcal{Y}$ (e.g., the sum of marginals of $P_0$ and $P_1$) such that the joint distributions can be described by densities $p_i(x, y)$ with respect to the product of the counting measure on $\mathcal{X}$ and $\nu$. Let $p_i(y) = \sum_{x \in \mathcal{X}} p_i(x, y)$ denote the marginal density of $Y$.

The conditional entropy can be written as:

$$H(X|Y) = \sum_{x \in \mathcal{X}} \int p(x, y) \log \frac{p(y)}{p(x, y)} \, d\nu(y).$$

Consider the function $f(u, v) = u \log(v/u)$ for $u, v \geq 0$. This function is the perspective transform of the concave function $g(t) = -t \log t$, given by $f(u, v) = vg(u/v)$. Since the perspective of a concave function is jointly concave, $f(u, v)$ is jointly concave in $(u, v)$.

For the mixture $P_\lambda$, the densities are linear combinations:

$$p_\lambda(x, y) = \lambda p_1(x, y) + (1 - \lambda)p_0(x, y), \quad p_\lambda(y) = \lambda p_1(y) + (1 - \lambda)p_0(y).$$

By the joint concavity of $f$:

$$f(p_\lambda(x, y), p_\lambda(y)) \geq \lambda f(p_1(x, y), p_1(y)) + (1 - \lambda)f(p_0(x, y), p_0(y)).$$

Summing over $x$ and integrating over $y$ preserves this inequality, yielding:

$$H_{P_\lambda}(X|Y) \geq \lambda H_{P_1}(X|Y) + (1 - \lambda)H_{P_0}(X|Y).$$

This proves the concavity. $\square$

Now we restate and prove Lemma 3.4.

**Lemma B.2.** *Let $\mathcal{M}$ be a membership tester for key size $n$ in universe $[u]$ with $u > n$. Let $p = \frac{n}{u}$ and let $F_p$ be as defined above. Then, the memory cost of $\mathcal{M}$ is at least:*

$$\frac{B(\mathcal{M})}{n} \geq F_p\big(\mu_K(\mathcal{M}), \mu_N(\mathcal{M})\big) - \frac{\log(8n)}{2n}.$$

*Proof.* We will apply the data processing inequality to lower bound the entropy of $W$ as a random variable, hence lower-bounding the memory usage of $\mathcal{M}$. Let $\mathcal{K}$ be uniformly randomly sampled from all subsets of $\mathcal{U} = [u]$ of size $n$. Define random variables $X_i = \mathbb{1}\{i \in \mathcal{K}\}$ to be the membership status for $i \in \mathcal{U}$, and define

$$\hat{X}_i = \mathsf{Query}^{\mathcal{M}}(i, \mathsf{Init}^{\mathcal{M}}(\mathcal{K}))$$

to be the answer to the membership query, treated as the "reconstructed" membership information. Then, by the data processing inequality we have

$$B(\mathcal{M}) = I(\mathcal{K}; W) \geq I(X^u; \hat{X}^u),$$

where $X^u$ and $\hat{X}^u$ denote the length-$u$ vector of $X_i$'s and $\hat{X}_i$'s, respectively. Because $X^u$ is discrete, we can write the mutual information as $I(X^u; \hat{X}^u) = H(X^u) - H(X^u | \hat{X}^u)$.

Let $I \sim \text{Unif}([u])$ be a uniformly random element, then by definition we have

$$I(X_I; \hat{X}_I) = p \cdot F_p\big(\mu_K(\mathcal{M}), \mu_N(\mathcal{M})\big).$$

We now relate the block mutual information $I(X^u; \hat{X}^u)$ to the single-letter mutual information $I(X_I; \hat{X}_I)$. First, by the chain rule for entropy and the fact that conditioning reduces entropy:

$$H(X^u | \hat{X}^u) = \sum_{i=1}^{u} H(X_i | \hat{X}^u, X_1, \ldots, X_{i-1})$$

$$\leq \sum_{i=1}^{u} H(X_i | \hat{X}_i).$$

To lower bound the average conditional entropy $\frac{1}{u} \sum_{i=1}^{u} H(X_i | \hat{X}_i)$, we use the fact that conditional entropy $H(X|Y)$ is a concave function in the joint distribution of $(X, Y)$. Since the distribution of $(X_I, \hat{X}_I)$ is the average of the distributions of $(X_i, \hat{X}_i)$, Jensen's inequality gives:

$$H(X_I | \hat{X}_I) \geq \frac{1}{u} \sum_{i=1}^{u} H(X_i | \hat{X}_i).$$

Combining the inequalities, we obtain the following upper bound:

$$H(X^u | \hat{X}^u) \leq \sum_{i=1}^{u} H(X_i | \hat{X}_i) \leq u H(X_I | \hat{X}_I).$$

Now, we lower bound the entropy of the source $H(X^u) = H(S) = \log \binom{u}{n}$. By standard approximations (see e.g. section 14.2 of (Cover & Thomas, 2001)), we have $\binom{u}{n} \geq \sqrt{\frac{u}{8n(u-n)}} 2^{uH(p)}$, so

$$H(X^u) = \log \binom{u}{n}$$

$$\geq u H(p) + \frac{1}{2} \log \frac{u}{8n(u-n)}$$

$$\geq u H(p) - \frac{1}{2} \log(8n).$$

Combining these results:

$$B(\mathcal{M}) \geq I(X^u; \hat{X}^u) = H(X^u) - H(X^u | \hat{X}^u)$$

$$\geq (u H(p) - \frac{1}{2} \log(8n)) - u H(X_I | \hat{X}_I)$$

$$= u(H(p) - H(X_I | \hat{X}_I)) - \frac{1}{2} \log(8n)$$

$$= u I(X_I; \hat{X}_I) - \frac{1}{2} \log(8n).$$

Since $u I(X_I; \hat{X}_I) = n \cdot F_p\big(\mu_K(\mathcal{M}), \mu_N(\mathcal{M})\big)$, it follows that:

$$\frac{B(\mathcal{M})}{n} \geq F_p\big(\mu_K(\mathcal{M}), \mu_N(\mathcal{M})\big) - \frac{\log(8n)}{2n},$$

as desired. $\qquad\qquad\qquad\qquad\qquad\qquad\qquad\qquad\qquad\qquad\qquad\qquad\qquad\qquad\qquad\qquad\qquad\qquad\quad\square$

### B.2. Proof of Lemma 3.5

**Lemma B.3** (Same as Lemma 3.5). *For $p > 0$, let $F_p(\mu_K, \mu_N) = I(X; \hat{X})/p$, where $X \sim \mathrm{Bern}(p)$, $\hat{X} \mid (X = 1) \sim \mu_K$, and $\hat{X} \mid (X = 0) \sim \mu_N$. We also define $F_0(\mu_K, \mu_N) = \mathrm{KL}(\mu_K \| \mu_N)$. Then, for $p \in [0, 1)$ and $(\mu_K, \mu_N) \in \mathcal{P}([0, 1])^2$:*

1. *The function $(p, \mu_K, \mu_N) \mapsto F_p(\mu_K, \mu_N)$ is jointly lower semi-continuous.*

2. *$F_p(\mu_K, \mu_N)$ is continuous in $p$.*

3. *$F_p$ is differentiable in $p$ with $\frac{\partial}{\partial p} F_p(\mu_K, \mu_N) = -\frac{\mathrm{KL}(\mu_N \| p\mu_K + (1-p)\mu_N)}{p^2}$ whenever $p > 0$.*

*Proof.* We prove the three properties using the following identity:

$$
\begin{aligned}
I(X; \hat{X}) &= \mathrm{KL}(P_{X, \hat{X}} \| P_X \otimes P_{\hat{X}}) \\
&= \mathop{\mathbb{E}}_{X \sim \mathrm{Bern}(p)} \mathrm{KL}(P_{\hat{X}|X} \| P_{\hat{X}}) \\
&= p\mathrm{KL}(\mu_K \| \mu_p) + (1-p)\mathrm{KL}(\mu_N \| \mu_p),
\end{aligned}
\tag{2}
$$

where $\mu_p = p\mu_K + (1-p)\mu_N$ is the marginal distribution of $\hat{X}$. Dividing by $p$, we obtain the expression for $F_p$:

$$
F_p(\mu_K, \mu_N) = \mathrm{KL}(\mu_K \| \mu_p) + \frac{1-p}{p}\mathrm{KL}(\mu_N \| \mu_p).
\tag{3}
$$

**1. Joint Lower Semi-Continuity.** For any sequence $(p_n, \mu_{K,n}, \mu_{N,n}) \to (p, \mu_K, \mu_N)$, we show that $\liminf F_{p_n} \geq F_p$. First, observe that the map $(p, \mu_K, \mu_N) \mapsto \mu_p = p\mu_K + (1-p)\mu_N$ is continuous from the product topology to the weak-* topology. Since KL divergence is jointly lower semi-continuous (LSC) with respect to the weak-* topology, it is immediate that $F_p(\mu_K, \mu_N)$ is LSC whenever $p > 0$.

Consider the case where $p = 0$. As $p_n \to 0$, $\mu_{p_n} \to \mu_N$, the first term converges like $\liminf \mathrm{KL}(\mu_{K,n} \| \mu_{p_n}) \geq \mathrm{KL}(\mu_K \| \mu_N)$. The second term $\frac{1-p_n}{p_n}\mathrm{KL}(\mu_{N,n} \| \mu_{p_n})$ is always non-negative. Therefore,

$$
\liminf F_{p_n} \geq \mathrm{KL}(\mu_K \| \mu_N) + 0 = F_0(\mu_K, \mu_N),
$$

proving its joint LSC everywhere.

**2. Continuity in $p$.** For fixed $\mu_K, \mu_N$, and $p > 0$, $F_p$ is clearly continuous as a composition of continuous functions. We only need to verify continuity at $p = 0$. Standard results in information theory have shown that the KL divergence vanishes sublinearly when the two distributions are close, see e.g. Proposition 2.20 in (Polyanskiy & Wu, 2025):

$$
\frac{\partial}{\partial p} \mathrm{KL}(\mu_N \| \mu_p) \bigg|_{p=0} = 0 \text{ iff } \mu_K \ll \mu_N.
$$

Hence, when $\mathrm{KL}(\mu_K \| \mu_N) < \infty$ (and hence $\mu_K \ll \mu_N$), we have

$$
\begin{aligned}
\lim_{p \to 0} F_p(\mu_K, \mu_N) &= \lim_{p \to 0} \left( \mathrm{KL}(\mu_K \| \mu_p) + \frac{1-p}{p}\mathrm{KL}(\mu_N \| \mu_p) \right) \\
&= \mathrm{KL}(\mu_K \| \mu_N) + \lim_{p \to 0} \frac{1-p}{p}\mathrm{KL}(\mu_N \| \mu_p) \\
&= \mathrm{KL}(\mu_K \| \mu_N),
\end{aligned}
$$

where the second equality is because the limit of first term is at least $\mathrm{KL}(\mu_K \| \mu_N)$ by LSC, and at most $\lim_{p \to 0}(1 - p)\mathrm{KL}(\mu_K \| \mu_N)$ by convexity. The last equality is because the second term vanishes by L'Hopital's rule.

When $\mathrm{KL}(\mu_K \| \mu_N) = \infty$, the limit also goes to infinity because $\mathrm{KL}(\mu_K \| \mu_p) \to \infty$.

**3. Differentiability in $p$.** Fix $\mu_K, \mu_N$. For $p \in (0,1)$, let $\nu := \mu_K + \mu_N$ and write $f_K := \frac{d\mu_K}{d\nu}$, $f_N := \frac{d\mu_N}{d\nu}$, so that the mixture $\mu_p = p\mu_K + (1-p)\mu_N$ has density

$$m_p := \frac{d\mu_p}{d\nu} = pf_K + (1-p)f_N.$$

Using (2), we can write

$$I(p) := I(X; \hat{X}) = p \int f_K \log \frac{f_K}{m_p} \, d\nu + (1-p) \int f_N \log \frac{f_N}{m_p} \, d\nu.$$

We first show that $I'(p) = \mathrm{KL}(\mu_K \| \mu_p) - \mathrm{KL}(\mu_N \| \mu_p)$. For convenience, we assume that the KL divergence is base-$e$ in this part and drop the $\ln 2$ factor. All results hold for base-2 as well.

For $p \in (0,1)$ we have

$$\frac{\partial}{\partial p} \log m_p = \frac{f_K - f_N}{m_p}, \qquad \text{and} \qquad \left| \frac{f_K - f_N}{m_p} \right| \leq \frac{f_K + f_N}{pf_K + (1-p)f_N} \leq \frac{1}{p} + \frac{1}{1-p}.$$

Therefore the functions $f_K \cdot \left| \frac{\partial}{\partial p} \log m_p \right|$ and $f_N \cdot \left| \frac{\partial}{\partial p} \log m_p \right|$ are $\nu$-integrable (bounded by constants times $f_K$ and $f_N$, which integrate to 1), and by dominated convergence we may differentiate $I(p)$ by moving $\frac{\partial}{\partial p}$ inside the integrals.

Differentiating $I(p)$ yields

$$I'(p) = \int f_K \log \frac{f_K}{m_p} \, d\nu - \int f_N \log \frac{f_N}{m_p} \, d\nu$$
$$+ p \int f_K \left( -\frac{\partial}{\partial p} \log m_p \right) d\nu + (1-p) \int f_N \left( -\frac{\partial}{\partial p} \log m_p \right) d\nu.$$

The last two terms cancel:

$$p \int f_K \left( -\frac{f_K - f_N}{m_p} \right) d\nu + (1-p) \int f_N \left( -\frac{f_K - f_N}{m_p} \right) d\nu = -\int (pf_K + (1-p)f_N) \frac{f_K - f_N}{m_p} \, d\nu$$
$$= -\int (f_K - f_N) \, d\nu = 0,$$

since $\int f_K \, d\nu = \int f_N \, d\nu = 1$. Hence,

$$I'(p) = \mathrm{KL}(\mu_K \| \mu_p) - \mathrm{KL}(\mu_N \| \mu_p).$$

Finally, $F_p = I(p)/p$ for $p > 0$, so by the quotient rule

$$\frac{\partial}{\partial p} F_p = \frac{pI'(p) - I(p)}{p^2}$$
$$= \frac{p\big(\mathrm{KL}(\mu_K \| \mu_p) - \mathrm{KL}(\mu_N \| \mu_p)\big) - \big(p\mathrm{KL}(\mu_K \| \mu_p) + (1-p)\mathrm{KL}(\mu_N \| \mu_p)\big)}{p^2}$$
$$= -\frac{\mathrm{KL}(\mu_N \| \mu_p)}{p^2}.$$

$\square$

### B.3. Proof for achievability of rate-distortion lower bound

Before proving Lemma 3.7, we need the following reduction, which states that for each $p$, it suffices to consider $\hat{x}$ as having finite support. The exact support set depends on $p$ and the error constraints, yet the size of this set is uniformly upper bounded by 5, by an application of Caratheodory's theorem.

**Lemma B.4** (Finite Support Reduction). *Fix $d^K, d^N, \varepsilon_K, \varepsilon_N > 0$ and $p \in (0,1)$. Then, there exists $(\mu_K^*, \mu_N^*) \in \mathcal{C}_K(\varepsilon_K) \times \mathcal{C}_N(\varepsilon_N)$ which attains the minimum in the definition of $R_p$,*

$$R_p(\varepsilon_K, \varepsilon_N) = F_p(\mu_K^*, \mu_N^*),$$

*and that $\mu_K^*, \mu_N^*$ are discrete distributions with $|\mathrm{supp}(\mu_K^*) \cup \mathrm{supp}(\mu_N^*)| \le 5$.*

*Proof.* We will start with any pair $(\mu_K^*, \mu_N^*)$ with errors exactly equal to $\varepsilon_K$ and $\varepsilon_N$ and attains the minimum in the definition of $R_p$. Then, we show that there exists some other $(\mu_K', \mu_N')$ which takes value on at most 5 points, achieves the same error levels, and attains the same rate:

$$R_p(\varepsilon_K, \varepsilon_N) = F_p(\mu_K^*, \mu_N^*) = F_p(\mu_K', \mu_N').$$

Consider the joint distribution $P_{X,\hat{X}}$ of $(X, \hat{X})$ that $(\mu_K^*, \mu_N^*)$ induces, and let $P_{\hat{X}} = p\mu_K^* + (1-p)\mu_N^*$ be the marginal distribution of $\hat{X}$. Then, for $P_{\hat{X}}$-almost every $\hat{x} \in [0,1]$, we have a unique posterior distribution for $X$, denoted $P_{X|\hat{X}}(\cdot|\hat{x}) = \mathrm{Bern}(q_{\hat{x}})$. The value $q_{\hat{x}}$ is characterized by the Radon-Nikodym derivative

$$q_{\hat{x}} = p \cdot \frac{d\mu_K^*}{dP_{\hat{X}}}(\hat{x}) \in [0,1].$$

For each $q \in [0,1]$, we use $h(q)$ to denote the binary entropy function $h(q) = -q\log_2 q - (1-q)\log_2(1-q)$, where $h(0) = h(1) = 0$. We now consider the $P_{\hat{X}}$-everywhere defined vector-valued function $f : E \to \mathbb{R}^4$, which maps each $\hat{x}$ to the tuple

$$f(\hat{x}) = \big(q_{\hat{x}},\ q_{\hat{x}} \cdot d^K(\hat{x}),\ (1 - q_{\hat{x}}) \cdot d^N(\hat{x}),\ h(q_{\hat{x}})\big).$$

Let $E$ be defined as the points over which $f$ is finite, and for convenience we will assume $(\mu_K^*, \mu_N^*)$ are tight for both error constraints. Although $d^K, d^N$ can take infinite values, $E$ has full $P_{\hat{X}}$-measure, as the error constraints imply that the following integrals are finite:

$$\mathop{\mathbb{E}}_{P_{X,\hat{X}}}[d^K(\hat{X})|X=1] = \frac{1}{p}\mathop{\mathbb{E}}_{P_{\hat{X}}}[q_{\hat{X}} \cdot d^K(\hat{X})] = \varepsilon_K \quad \text{and} \quad \mathop{\mathbb{E}}_{P_{X,\hat{X}}}[d^N(\hat{X})|X=0] = \frac{1}{1-p}\mathop{\mathbb{E}}_{P_{\hat{X}}}[(1 - q_{\hat{X}}) \cdot d^N(\hat{X})] = \varepsilon_N.$$

Moreover, we also note that by construction,

$$\mathop{\mathbb{E}}_{\hat{X} \sim P_{\hat{X}}}[q_{\hat{X}}] = p \quad \text{and} \quad \mathop{\mathbb{E}}_{\hat{X} \sim P_{\hat{X}}}[h(q_{\hat{X}})] = H(X|\hat{X}) = h(p) - p \cdot F_p(\mu_K^*, \mu_N^*).$$

Note that $\mathbb{E}_{\hat{X} \sim P_{\hat{X}}}[f(\hat{X})] = \big(p, p\varepsilon_K, (1-p)\varepsilon_N, h(p) - p \cdot F_p(\mu_K^*, \mu_N^*)\big)$ is in the convex hull of the vectors

$$\big\{(q_{\hat{x}},\ q_{\hat{x}} \cdot d^K(\hat{x}),\ (1 - q_{\hat{x}}) \cdot d^N(\hat{x}),\ h(q_{\hat{x}})) : \hat{x} \in E\big\}.$$

By Carathéodory's Theorem, there exist $\alpha_1, \ldots, \alpha_5 \ge 0$ and $\hat{x}_1, \ldots, \hat{x}_5 \in [0,1]$ such that $\sum_{i=1}^5 \alpha_i = 1$ and

$$\mathop{\mathbb{E}}_{\hat{X} \sim P_{\hat{X}}}[f(\hat{X})] = \sum_{i=1}^5 \alpha_i f(\hat{x}_i).$$

Now, consider the random variable $X' \sim \sum_{i=1}^5 \alpha_i \delta_{\hat{x}_i}$, where $\delta_{\hat{x}_i}$ is the Dirac delta distribution at $\hat{x}_i$. Let $P_{X|X'}$ be defined as $P_{X|X'}(x|x') = \mathrm{Bern}(q_{x'})$. We can check that $\mathbb{P}[X = 1] = p$ under this $P_{X|X'}$ since $\sum_{i=1}^5 \alpha_i q_{\hat{x}_i} = p$. We can now construct $\mu_K'$ and $\mu_N'$ as

$$\mu_K' = \frac{1}{p}\sum_{i=1}^5 \alpha_i q_{\hat{x}_i} \delta_{\hat{x}_i} \quad \text{and} \quad \mu_N' = \frac{1}{1-p}\sum_{i=1}^5 \alpha_i(1 - q_{\hat{x}_i})\delta_{\hat{x}_i}.$$

We have $\mu'_K \in \mathcal{C}_K(\varepsilon_K)$ and $\mu'_N \in \mathcal{C}_N(\varepsilon_N)$ since

$$\underset{X' \sim \mu'_K}{\mathbb{E}}[d^K(X')] = \frac{1}{p}\sum_{i=1}^{5}\alpha_i q_{\hat{x}_i} \cdot d^K(\hat{x}_i) = \varepsilon_K$$

and

$$\underset{X' \sim \mu'_N}{\mathbb{E}}[d^N(X')] = \frac{1}{1-p}\sum_{i=1}^{5}\alpha_i(1 - q_{\hat{x}_i}) \cdot d^N(\hat{x}_i) = \varepsilon_N.$$

Finally, $(\mu'_K, \mu'_N)$ also achieve the same rate as $(\mu^*_K, \mu^*_N)$ since

$$p \cdot F_p(\mu'_K, \mu'_N) = h(p) - \underset{X' \sim P_{X'}}{\mathbb{E}}[h(q_{X'})] = h(p) - \underset{\hat{X} \sim P_{\hat{X}}}{\mathbb{E}}[h(q_{\hat{X}})] = p \cdot F_p(\mu^*_K, \mu^*_N).$$

It follows that $(\mu'_K, \mu'_N)$ attains the minimum in the definition of $R_p$ and has support on at most 5 points. $\qquad\square$

Now that we reduced the reconstructed $\hat{X}$ to a random variable with finite support, we can proceed with the proof of achievability.

**Lemma B.5** (same as Lemma 3.7). *Fix any $p = \frac{n}{u} \in (0,1)$ and $\varepsilon_K, \varepsilon_N > 0$. Then, for all $\delta > 0$, there is a sufficiently large $n$ and $u$ such that there exists a membership tester $\mathcal{M}$ for universe $[u]$ and key sizes $n$ which achieves error rates $(\varepsilon_K + \delta, \varepsilon_N + \delta)$, and:*

$$\frac{B(\mathcal{M})}{n} \leq R_p(\varepsilon_K, \varepsilon_N) + \delta.$$

*Proof.* **Step 1: Finite Support and Continuity.** Let $(\mu^*_K, \mu^*_N)$ be the pair of distributions attaining the minimum in the definition of $R_p(\varepsilon_K, \varepsilon_N)$. By the finite support reduction (proven in the previous lemma), we assume without loss of generality that these distributions are supported on a finite set $\mathcal{Y}^* \subset [0,1]$ with $|\mathcal{Y}^*| \leq 5$. In the rest of the proof, we will rename the recovered $\hat{X}$ as $Y$ for notational convenience.

Let $P^*_{X,Y}$ be the joint distribution of $X$ and $Y$ under the optimal $(\mu^*_K, \mu^*_N)$, where $P^*_X = \mathrm{Bern}(p)$ is the source distribution, and $P^*_{Y|X}$ is the transition kernel defined by $P^*_{Y|X}(y|1) = \mu^*_K(y)$ and $P^*_{Y|X}(y|0) = \mu^*_N(y)$.

We will use $\mathcal{D}$ to denote the set of joint distributions $P_{X,Y}$ with $\mathrm{Bern}(p)$ marginal on $X$ and which places zero mass on points with infinite errors: for all $y \in \mathcal{Y}^*$, $P_{X,Y}(1,y) = 0$ whenever $d^K(y) = \infty$ and $P_{X,Y}(0,y) = 0$ whenever $d^N(y) = \infty$. Since both $X$ and $Y$ have finite support, $\mathcal{D}$ is compact. Consequently, the following functionals are uniformly continuous over $\mathcal{D}$:

1. $D_K(P_{X,Y}) = \mathbb{E}_{y \sim P_{Y|X}(\cdot|1)}[d^K(y)]$;

2. $D_N(P_{X,Y}) = \mathbb{E}_{y \sim P_{Y|X}(\cdot|0)}[d^N(y)]$.

Therefore, there exists a $\gamma > 0$ such that for any $P'_{X,Y} \in \mathcal{D}$ with $\|P'_{X,Y} - P^*_{X,Y}\|_1 < \gamma$, we have:

$$|D_K(P'_{X,Y}) - D_K(P^*_{X,Y})| < \delta, \tag{4}$$
$$|D_N(P'_{X,Y}) - D_N(P^*_{X,Y})| < \delta, \tag{5}$$

**Step 2: Typical Sequences.** For each $P_{X,Y} \in \mathcal{D}$, and for each source sequence $x^u \in \{0,1\}^u$, let $T^u_{[P_{Y|X}]\gamma}(x^u)$ be the set of $P_{Y|X}$-*typical sequences under condition $x^u$ with constant $\gamma$* as defined in Definition 2.9 of Csiszár & Körner (2011). These sequences are $y^u \in \mathcal{Y}^{*u}$ such that:

1. Let $\hat{P}_{X,Y}(x', y'|x^u, y^u)$ be the joint empirical distribution of $x_i$ and $y_i$ for $i = 1, \ldots, u$ in the sequences $x^u$ and $y^u$. Then,

$$\left|\hat{P}_{X,Y}(x', y'|x^u, y^u) - P_X(x')P_{Y|X}(y'|x')\right| \leq \gamma.$$

2. Whenever $P_{Y|X}(y'|x') = 0$, $\hat{P}_{X,Y}(x', y'|x^u, y^u) = 0$.

It follows from uniform continuity that, for sufficiently small $\gamma$, all $P_{Y|X}^*$-typical sequences $y^u$ under condition $x^u$ will have the empirical distribution of $(x_i, y_i)$ close to $P_{X,Y}^*$, and thus satisfy (4) and (5) for the desired choice of $\delta$.

By Lemma 2.13 in Csiszár & Körner (2011), there exist sequences $\delta_u \to 0$ and $\gamma_u \to 0$ such that for all $x^u \in \{0,1\}^u$ with $n = pu$ entries of 1,

$$\left| \frac{1}{u} \log |T_{[P_{Y|X}]\gamma_u}^u (x^u)| - H(Y|X) \right| \leq \delta_u,$$

where $H(Y|X)$ is based on the selected $P_{X,Y}$.

Similarly, we can define $P_Y$-*typical sequences with constant* $\gamma$ as the sequences $y^u \in \mathcal{Y}^u$ such that

$$\left| \hat{P}_Y(y'|y^u) - P_Y(y') \right| \leq \gamma,$$

where $\hat{P}_Y(y'|y^u)$ is the empirical distribution of $y'$ in $y^u$. There exist sequences $\delta_u \to 0$ and $\gamma_u \to 0$ such that

$$\left| \frac{1}{u} \log |T_{[P_Y]\gamma_u}^u| - H(Y) \right| \leq \delta_u,$$

where $H(Y)$ is based on the selected $P_Y$.

**Step 3: Type Covering.** By the previous steps, for sufficiently large $u$, there is some $\gamma$ such that for each source sequence $x^u$ with $n$ entries of 1, each $y \in T_{[P_{Y|X}]\gamma}^u(x^u)$ satisfies (4) and (5) for the desired choice of $\delta$. Now, we apply the same proof as the type covering lemma (Lemma 9.1 in Csiszár & Körner (2011)) to show that all valid source $x^u$ can be "covered" by some $y^u$ in a fixed codebook $M$. We will demonstrate the existence of such a codebook and bound its size.

Let $P_Y^*$ be the marginal distribution of $Y$ under $P_{X,Y}^*$. Consider a codebook $M$ of $m$ sequences sampled uniformly from $T_{[P_Y^*]\gamma}^u$, and let $Y^u$ be such a random sequence. For any source sequence $x^u$ with $n$ entries of 1, the probability that $Y^u$ is $P_{X|Y}^*$-typical under condition $x^u$ with constant $\gamma$ is given by

$$\mathbb{P}[Y^u \in T_{[P_{Y|X}^*]\gamma}^u(x^u)] = \frac{|T_{[P_{Y|X}^*]\gamma}^u(x^u)|}{|T_{[P_Y^*]\gamma}^u|} \geq \frac{2^{-u(H(Y|X)+p\delta/4)}}{2^{u(H(Y)+p\delta/4)}} = 2^{-u(I(X;Y)+p\delta/2)},$$

where $X, Y$ are distributed according to $P_{X,Y}^*$. The inequality in the middle is due to the fact that $\delta_u$ is eventually smaller than $p\delta/4$, for our choice of $\delta$.

With $m$ uniformly selected $Y^u$, the expected number of $x^u$ that are *not* covered by any $Y^u$ in the codebook is at most:

$$\mathbb{E}[\text{number of } x^u \text{ not covered by } M] = \sum_{x^u} \mathbb{P}[Y^u \notin T_{[P_{Y|X}^*]\gamma}^u(x^u)]$$

$$\leq \sum_{x^u} \left(1 - 2^{-u(I(X;Y)+p\delta/2)}\right)^m$$

$$\leq \exp(-m 2^{-u(I(X;Y)+p\delta/2)}) \cdot \binom{u}{n}.$$

For sufficiently large $u$, this value is strictly less than 1 if we choose $m = 2^{u(I(X;Y)+p\delta)}$. As the expectation is smaller than 1, this means that there exists a choice of codebook that simultaneously covers all source sequences.

We can now define a membership tester, which initializes by encoding the nearest codeword in $M$ to the $x^u$ corresponding to key set $\mathcal{K}$, and on query $i$, outputs the $i$th entry of that codeword. And the proof is finished since the memory cost is at most $\log |M| \leq u(I(X;Y) + p\delta) = n \cdot (R_p(\varepsilon_K, \varepsilon_N) + \delta)$. □

### B.4. Proof of Theorem 3.1

**Theorem B.6** (same as Theorem 3.1). *Fix error metrics $d^K, d^N$ and error rates $\varepsilon_K, \varepsilon_N \geq 0$. Let $\{n_j\}, \{u_j\}$ be sequences of natural numbers such that $n_j \to \infty$ and $n_j/u_j \to 0$. For each $j$, let $\mathcal{M}_j$ be a membership tester for universe $[u_j]$ and*

*key size $n_j$ that achieves error rates $(\varepsilon_{K,j}, \varepsilon_{N,j})$ under error metrics $d^K, d^N$. Suppose the error rates satisfy:*

$$\limsup_{j \to \infty} \varepsilon_{K,j} \leq \varepsilon_K, \quad \limsup_{j \to \infty} \varepsilon_{N,j} \leq \varepsilon_N.$$

*Then, the asymptotic per-key memory budget of $\mathcal{M}_j$ is at least:*

$$\liminf_{j \to \infty} \frac{B(\mathcal{M}_j)}{n_j} \geq \min_{\mu_K \in \mathcal{C}_K(\varepsilon_K), \mu_N \in \mathcal{C}_N(\varepsilon_N)} \mathrm{KL}(\mu_K \| \mu_N).$$

*Moreover, there exist sequences $\{u_j\}$, $\{n_j\}$, and $\{\mathcal{M}_j\}$ as described above that achieve the memory lower bound.*

Now, Theorem 3.1 follows from the fact that $R_p$ converges to the minimum KL divergence as $p \to 0$.

*Proof of Theorem 3.1.* First we show the lower bound. By Lemma 3.4, for each $j$, the memory cost of $\mathcal{M}_j$ satisfies:

$$\frac{B(\mathcal{M}_j)}{n_j} \geq F_{p_j}\big(\mu_K(\mathcal{M}_j), \mu_N(\mathcal{M}_j)\big) - \frac{\log(8n_j)}{2n_j}.$$

Let $\ell = \liminf_{j \to \infty} \frac{B(\mathcal{M}_j)}{n_j}$. By passing to a subsequence (still denoted by $j$), we can assume $B(\mathcal{M}_j)/n_j \to \ell$.

Let $\mu_{K,j} = \mu_K(\mathcal{M}_j)$ and $\mu_{N,j} = \mu_N(\mathcal{M}_j)$. Since the space of probability measures $\mathcal{P}([0,1])$ is compact under the weak-* topology, there exists a further subsequence (again, denoted by $j$) such that $(\mu_{K,j}, \mu_{N,j})$ converges to some limit $(\mu_K^*, \mu_N^*) \in \mathcal{P}([0,1])^2$. By the lower semi-continuity of $F_p(\mu_K, \mu_N)$ in $(p, \mu_K, \mu_N)$ (Lemma 3.5), and since $p_j \to 0$, we have:

$$\ell = \lim_{j \to \infty} \frac{B(\mathcal{M}_j)}{n_j} \geq \liminf_{j \to \infty} F_{p_j}(\mu_{K,j}, \mu_{N,j})$$
$$\geq F_0(\mu_K^*, \mu_N^*) = \mathrm{KL}(\mu_K^* \| \mu_N^*).$$

It remains to show that $(\mu_K^*, \mu_N^*)$ is in $\mathcal{C}_K(\varepsilon_K) \times \mathcal{C}_N(\varepsilon_N)$. By assumption, we have $\limsup \varepsilon_{K,j} \leq \varepsilon_K$ and $\limsup \varepsilon_{N,j} \leq \varepsilon_N$. Since the functions $d^K, d^N$ are LSC, the expected error is lower semi-continuous with respect to the measure by Portmanteau theorem. Thus:

$$\mathbb{E}_{\hat{x} \sim \mu_K^*}[d^K(\hat{x})] \leq \liminf_{j \to \infty} \mathbb{E}_{\hat{x} \sim \mu_{K,j}}[d^K(\hat{x})] \leq \varepsilon_K,$$

which implies $\mu_K^* \in \mathcal{C}_K(\varepsilon_K)$, and similarly $\mu_N^* \in \mathcal{C}_N(\varepsilon_N)$. This completes the lower bound:

$$\liminf_{j \to \infty} \frac{B(\mathcal{M}_j)}{n_j} \geq \mathrm{KL}(\mu_K^* \| \mu_N^*)$$
$$\geq \min_{\mu_K \in \mathcal{C}_K(\varepsilon_K), \mu_N \in \mathcal{C}_N(\varepsilon_N)} \mathrm{KL}(\mu_K \| \mu_N).$$

For achievability, let $(\bar{\mu}_K, \bar{\mu}_N)$ be any minimizer of the KL objective over $\mathcal{C}_K(\varepsilon_K) \times \mathcal{C}_N(\varepsilon_N)$. Since $F_p(\bar{\mu}_K, \bar{\mu}_N) \to \mathrm{KL}(\bar{\mu}_K \| \bar{\mu}_N)$ as $p \to 0$ by Lemma 3.5, we also have

$$R_p(\varepsilon_K, \varepsilon_N) \leq F_p(\bar{\mu}_K, \bar{\mu}_N) \to \min_{\mu_K \in \mathcal{C}_K(\varepsilon_K), \mu_N \in \mathcal{C}_N(\varepsilon_N)} \mathrm{KL}(\mu_K \| \mu_N).$$

By Lemma 3.7, for each $p$, for sufficiently large $n$ we can find a membership tester with error rates arbitrarily close to $(\varepsilon_K, \varepsilon_N)$ and memory usage arbitrarily close to $R_p(\varepsilon_K, \varepsilon_N)$ per key. The desired claim follows by taking $p_j \to 0$ and taking a sufficiently large $n_j$ for each $p_j$. $\square$

## B.5. Proof of Theorem 3.2

**Theorem B.7** (same as Theorem 3.2). *Suppose $(\mu_K^*, \mu_N^*) \in \mathcal{C}_K(\varepsilon_K) \times \mathcal{C}_N(\varepsilon_N)$ is the unique minimizer of $\mathrm{KL}(\mu_K \| \mu_N)$. In the setting of Theorem 3.1, if $\{\mathcal{M}_j\}$ is asymptotically optimal, in the sense that*

$$\limsup_{j \to \infty} \frac{B(\mathcal{M}_j)}{n_j} = \mathrm{KL}(\mu_K^* \| \mu_N^*),$$

*then we must have $\mu_K(\mathcal{M}_j) \to \mu_K^*$ and $\mu_N(\mathcal{M}_j) \to \mu_N^*$ in Wasserstein-1 distance.*

*Proof of Theorem 3.2.* Consider the output distributions $(\mu_K(\mathcal{M}_j), \mu_N(\mathcal{M}_j))$ for $j = 1, 2, \ldots$. By compactness of $\mathcal{P}([0,1])^2$, there exists a subsequence of $(\mu_K(\mathcal{M}_j), \mu_N(\mathcal{M}_j))$ converging to some $(\mu_K'^*, \mu_N'^*) \in \mathcal{P}([0,1])^2$.

As shown in Section B.4, the limit point $(\mu_K'^*, \mu_N'^*)$ must satisfy the error constraints, i.e., $(\mu_K'^*, \mu_N'^*) \in \mathcal{C}_K(\varepsilon_K) \times \mathcal{C}_N(\varepsilon_N)$. By the memory-optimality of $\{\mathcal{M}_j\}$, if $(\mu_K^*, \mu_N^*)$ are the unique minimizers of KL divergence, then we have:

$$
\begin{aligned}
\mathrm{KL}(\mu_K^* \| \mu_N^*) &= \limsup_{j \to \infty} \frac{B(\mathcal{M}_j)}{n_j} \\
&\geq \limsup_{j \to \infty} \left( F_{p_j}(\mu_K(\mathcal{M}_j), \mu_N(\mathcal{M}_j)) + O\left(\frac{\log n_j}{n_j}\right) \right) \\
&\geq \mathrm{KL}(\mu_K'^* \| \mu_N'^*).
\end{aligned}
$$

Hence $(\mu_K^*, \mu_N^*) = (\mu_K'^*, \mu_N'^*)$. Because any convergent subsequence of $(\mu_K(\mathcal{M}_j), \mu_N(\mathcal{M}_j))$ converges to $(\mu_K^*, \mu_N^*)$, and the space $\mathcal{P}([0,1])^2$ is compact, it follows that the whole sequence $(\mu_K(\mathcal{M}_j), \mu_N(\mathcal{M}_j))$ converges to $(\mu_K^*, \mu_N^*)$. The desired claim then follows from the fact that convergence in Wasserstein-1 is equivalent to weak-* convergence on $[0,1]$. $\square$

## B.6. Proof of Theorem 3.3

**Theorem B.8** (same as Theorem 3.3). *Let $\chi^2$ be the chi-squared divergence. In the setting of Theorem 3.2, suppose $\mu_N^*$ is supported on a finite set $\mathcal{X}$ and $\chi^2(\mu_K^* \| \mu_N^*) < \infty$. For any membership tester $\mathcal{M}$ for key size $n$ and universe $[u]$ with query outputs restricted to $\mathcal{X}$, if we fix $p = \frac{n}{u}$ and let $n, u \to \infty$, then*

$$
\frac{B(\mathcal{M})}{n} \geq \mathrm{KL}(\mu_K^* \| \mu_N^*) - \frac{\chi^2(\mu_K^* \| \mu_N^*)}{2 \ln 2} \cdot p + o(p),
$$

*and this bound is achievable.*

*Proof.* Let $\delta := \min_{x \in \mathcal{X}} \mu_N^*(x) > 0$. For $p \in (0, 1)$, recall

$$
R_p(\varepsilon_K, \varepsilon_N) = \min_{(\mu_K, \mu_N) \in \mathcal{C}} F_p(\mu_K, \mu_N), \qquad F_0(\mu_K, \mu_N) = \mathrm{KL}(\mu_K \| \mu_N),
$$

where $\mathcal{C} = \mathcal{C}_K(\varepsilon_K) \times \mathcal{C}_N(\varepsilon_N) \cap \mathcal{P}(\mathcal{X})$ and $\mathcal{P}(\mathcal{X}) \subseteq \mathbb{R}^{\mathcal{X}}$ is the set of all probability distributions on $\mathcal{X}$.

**Step 1: a fixed compact choice set for small $p$.** Let $(\mu_K(p), \mu_N(p)) \in \mathcal{C}$ be any mapping from $p$ to minimizers of $F_p$. As shown in the proof of Theorem 3.2, $\mu_N(p) \Rightarrow \mu_N^*$ as $p \to 0$, hence for all sufficiently small $p$ we have $\mu_N(p)(x) \geq \delta/2$ for every $x \in \mathcal{X}$. Therefore, for some $p_0 > 0$ we may restrict attention to the compact set

$$
\mathcal{C}' := \mathcal{C} \cap \left\{ (\mu_K, \mu_N) : \mu_N(x) \geq \delta/2 \,\forall x \in \mathcal{X} \right\},
$$

and for all $p \in (0, p_0]$,

$$
R_p(\varepsilon_K, \varepsilon_N) = \min_{(\mu_K, \mu_N) \in \mathcal{C}'} F_p(\mu_K, \mu_N),
$$

since there is a minimizer $(\mu_K(p), \mu_N(p))$ that lies in $\mathcal{C}'$ for $p$ sufficiently small.

**Step 2: verify assumptions for envelope theorem.** Define a maximization problem with value function

$$
V(p) := \max_{(\mu_K, \mu_N) \in \mathcal{C}'} f((\mu_K, \mu_N), p), \qquad f((\mu_K, \mu_N), p) := -F_p(\mu_K, \mu_N),
$$

so that $V(p) = -R_p(\varepsilon_K, \varepsilon_N)$ for all $p \in [0, p_0]$. Now, we verify the assumptions of Corollary 4 of Milgrom and Segal (Milgrom & Segal, 2002). We will use $f_p$ to denote $\frac{\partial f}{\partial p}$ in the rest of the proof.

- *Compactness.* $\mathcal{C}' \subset \mathbb{R}^{2|\mathcal{X}|}$ is closed and contained in a product of simplices, hence compact.

- *Continuity of $f$.* Since $\mu_N(x) \geq \delta/2$ on $\mathcal{C}'$ and $\mu_p := p\mu_K + (1-p)\mu_N$ satisfies $\mu_p(x) \geq (1-p_0)\delta/2 > 0$ for $p \in [0, p_0]$, all KL terms appearing in the expression of $F_p$ are finite and continuous; thus $(\mu_K, \mu_N, p) \mapsto f((\mu_K, \mu_N), p)$ is continuous on $\mathcal{C}' \times [0, p_0]$.

- *Continuity of $f_p$.* For $p > 0$, Lemma 3.5 gives

$$\frac{\partial}{\partial p} F_p(\mu_K, \mu_N) = -\frac{\mathrm{KL}(\mu_N \| p\mu_K + (1-p)\mu_N)}{p^2}, \qquad (p > 0),$$

hence

$$f_p((\mu_K, \mu_N), p) = \frac{\mathrm{KL}(\mu_N \| \mu_p)}{p^2}, \qquad (p > 0).$$

Fix $(\mu_K, \mu_N) \in \mathcal{C}'$ and write $h(x) := \frac{\mu_K(x)}{\mu_N(x)} - 1$. Then $h(x) \geq -1$ and, since $\mu_N(x) \geq \delta/2$ on $\mathcal{C}'$, we have a uniform bound

$$M := \sup_{(\mu_K, \mu_N) \in \mathcal{C}'} \|h\|_\infty < \infty.$$

Pick any constant $c \in (0, 1)$ and let $p_1 := \min\{p_0, c/M\}$. Then for all $p \in [0, p_1]$, all $(\mu_K, \mu_N) \in \mathcal{C}'$, and all $x \in \mathcal{X}$, we have $|ph(x)| \leq c$. By Taylor's theorem for $\log(1 + u)$ around $u = 0$ with a uniform remainder bound on the compact interval $u \in [-c, c]$, we obtain

$$-\log(1 + u) = -u + \frac{u^2}{2} + O(u^3) \qquad \text{uniformly for } |u| \leq c.$$

Therefore,

$$\mathrm{KL}_e(\mu_N \| \mu_p) = \sum_{x \in \mathcal{X}} \mu_N(x)\Big[-\log(1 + ph(x))\Big] = \frac{p^2}{2} \sum_{x \in \mathcal{X}} \mu_N(x)h(x)^2 + O(p^3),$$

where the $O(p^3)$ term is uniform over $(\mu_K, \mu_N) \in \mathcal{C}'$. Consequently,

$$\mathrm{KL}(\mu_N \| \mu_p) = \frac{1}{\ln 2} \mathrm{KL}_e(\mu_N \| \mu_p) = \frac{p^2}{2 \ln 2} \chi^2(\mu_K \| \mu_N) + O(p^3),$$

uniformly over $(\mu_K, \mu_N) \in \mathcal{C}'$. Hence $f_p$ extends continuously to $p = 0$ by

$$f_p((\mu_K, \mu_N), 0) := \frac{1}{2 \ln 2} \chi^2(\mu_K \| \mu_N),$$

and $f_p$ is continuous on $\mathcal{C}' \times [0, p_0]$.

**Step 3: apply envelope theorem and conclude the first-order expansion.** By Milgrom and Segal's Corollary 4(ii) (Milgrom & Segal, 2002), $V$ is absolutely continuous and its right derivative satisfies

$$V'_+(p) = \max_{(\mu_K, \mu_N) \in \arg\max_{\mathcal{C}'} f(\cdot, p)} f_p((\mu_K, \mu_N), p) \quad \text{for } p \in [0, p_0).$$

Since the optimizer at $p = 0$ is unique, $\arg\max_{\mathcal{C}'} f(\cdot, 0) = \{(\mu_K^*, \mu_N^*)\}$ and thus

$$V'_+(0) = f_p((\mu_K^*, \mu_N^*), 0) = \frac{1}{2 \ln 2} \chi^2(\mu_K^* \| \mu_N^*).$$

Since $R_p = -V(p)$, it follows that as $p \to 0$,

$$R_p(\varepsilon_K, \varepsilon_N) = \mathrm{KL}(\mu_K^* \| \mu_N^*) - \frac{\chi^2(\mu_K^* \| \mu_N^*)}{2 \ln 2} p + o(p).$$

$\square$

# C. Proofs for Section 4.1

Our proof is an application of Theorem 3.1. We solve the following convex optimization problem and show that $\mu_K^*$ and $\mu_N^*$ are the unique optimal solutions. We note that, since the base-$e$ KL divergence is a constant multiple of the base-2 KL divergence, we assume KL divergence is base-$e$ for convenience of analysis. The shape of optimal solutions is unaffected by this choice of base. In this appendix, we use $X$ to denote the score random variable (i.e., $\hat{X}$ in the main text).

$$\min_{\mu_K, \mu_N \in \mathcal{P}([0,1])} \mathrm{KL}(\mu_K \| \mu_N), \quad \text{subject to} \quad \left\{ \begin{array}{l} \mathbb{E}_{X \sim \mu_K}[-\ln X] \leq \varepsilon_K, \\ \mathbb{E}_{X \sim \mu_N}[-\ln(1-X)] \leq \varepsilon_N. \end{array} \right. \tag{6}$$

The Lagrangian is:

$$\mathcal{L}(\mu_K, \mu_N, \lambda_K, \lambda_N) = \mathrm{KL}(\mu_K \| \mu_N) + \lambda_K \big( \mathbb{E}_{\mu_K}[-\ln X] - \varepsilon_K \big) + \lambda_N \big( \mathbb{E}_{\mu_N}[-\ln(1-X)] - \varepsilon_N \big).$$

Before we proceed, we introduce the following technical lemmas for optimization over measure spaces.

## C.1. Technical preliminaries

To analyze the optimality condition for convex functionals over the space of measures, we use Gâteaux derivatives.

**Definition C.1** (Gâteaux Derivative). Let $J : \mathcal{P}([0,1]) \to \mathbb{R}$ be a linear functional. The Gâteaux derivative of $J$ at $Q \in \mathcal{P}([0,1])$ in the direction $R - Q$, where $R \in \mathcal{P}([0,1])$, is defined as:

$$\delta J(Q; R - Q) = \lim_{\eta \to 0^+} \frac{J((1-\eta)Q + \eta R) - J(Q)}{\eta},$$

provided the limit exists.

If $J$ is a convex functional, the Gâteaux derivative allows us to state the necessary and sufficient conditions for global optimality.

**Theorem C.2** (First-Order Optimality Condition; see e.g. (Luenberger, 1997)). *If $J : \mathcal{P}([0,1]) \to \mathbb{R}$ is convex and Gâteaux differentiable, then $Q^* \in \mathcal{P}([0,1])$ minimizes $J$ if and only if $\delta J(Q^*; R - Q^*) \geq 0$ for all $R \in \mathcal{P}([0,1])$.*

When the Gâteaux derivative can be represented in an integral form (and in particular linear in $R - Q$), this optimality condition translates into a structural property regarding the support of the optimal measure.

**Lemma C.3** (KKT Support Condition). *Suppose the Gâteaux derivative of a convex functional $J$ at $Q$ can be represented as*

$$\delta J(Q; R - Q) = \int_0^1 g_Q(x) dR(x) - \int_0^1 g_Q(x) dQ(x) = \mathbb{E}_R[g_Q(X)] - \mathbb{E}_Q[g_Q(X)],$$

*for some measurable function $g_Q(x)$ (which may depend on $Q$). Then $Q^*$ minimizes $J$ if and only if $Q^*$ is supported on the set of global minima of the function $g_{Q^*}(x)$.*

*That is, $x \in supp(Q^*) \implies g_{Q^*}(x) = \inf_{y \in [0,1]} g_{Q^*}(y)$.*

*Proof.* By Theorem C.2, $Q^*$ is optimal iff $\mathbb{E}_R[g_{Q^*}(X)] \geq \mathbb{E}_{Q^*}[g_{Q^*}(X)]$ for all $R$. Let $K^* = \mathbb{E}_{Q^*}[g_{Q^*}(X)]$. If we take $R = \delta_y$ for any $y \in [0,1]$, we get $g_{Q^*}(y) \geq K^*$. Thus $K^*$ is the global minimum value of $g_{Q^*}(x)$. Since $Q^*$ is a probability measure, the equality $\mathbb{E}_{Q^*}[g_{Q^*}(X)] = K^*$ can only hold if $Q^*$ is supported entirely on the set where $g_{Q^*}(x) = K^*$. $\square$

We also recall the following standard result regarding the minimization of KL divergence subject to linear constraints.

**Lemma C.4** (Donsker–Varadhan variational formula). *Suppose $Q \in \mathcal{P}([0,1])$, and let $h$ be a $Q$-measurable real function such that $\mathbb{E}_Q[e^{-h(X)}] < \infty$. Then,*

$$-\ln \mathbb{E}_{X \sim Q}[e^{-h(X)}] = \inf_{P \in \mathcal{P}([0,1])} \left\{ \mathrm{KL}(P \| Q) + \mathbb{E}_P[h(X)] \right\}.$$

*The infimum is attained when $P$ has the Radon-Nikodym derivative $\frac{dP}{dQ}(x) \propto e^{-h(x)}$.*

## C.2. Solving the optimization problem

**Step 1: stationary condition for $\mu_K$.**   Applying the variational formula Theorem C.4 with $h(x) = -\lambda_K \ln x$, the first two terms are minimized when the following inf is attained:

$$\inf_{\mu_K} \left\{ \mathrm{KL}(\mu_K \| \mu_N) + \mathbb{E}_{\mu_K}[\lambda_K(-\ln X)] \right\}.$$

By Theorem C.4, this is when

$$\frac{d\mu_K}{d\mu_N}(x) \propto e^{-h(x)} = x^{\lambda_K}. \tag{7}$$

Let $C(\mu_N) = \mathbb{E}_{\mu_N}[X^{\lambda_K}]$ be the normalization constant. The value of the infimum is $-\ln C(\mu_N)$.

**Step 2: stationary condition for $\mu_N$.**   Plugging the optimized $\mu_K$ back into the Lagrangian, we obtain the dual function which we seek to minimize over $\mu_N$:

$$J(\mu_N) = -\ln \mathbb{E}_{\mu_N}[X^{\lambda_K}] + \lambda_N \mathbb{E}_{\mu_N}[-\ln(1-X)] - \lambda_K \varepsilon_K - \lambda_N \varepsilon_N.$$

$J(\mu_N)$ is a convex functional of $\mu_N$. We use Gâteaux derivatives $\delta J(\mu_N^*; R - \mu_N^*)$ to find the optimality condition.

The derivative of the first term $J_1(\mu_N) = -\ln \mathbb{E}_{\mu_N}[X^{\lambda_K}]$ is:

$$\delta J_1(\mu_N; R - \mu_N) = -\frac{\mathbb{E}_R[X^{\lambda_K}] - \mathbb{E}_{\mu_N}[X^{\lambda_K}]}{\mathbb{E}_{\mu_N}[X^{\lambda_K}]} = -\frac{\mathbb{E}_R[X^{\lambda_K}] - \mathbb{E}_{\mu_N}[X^{\lambda_K}]}{C(\mu_N)}.$$

The derivative of the second (linear) term $J_2(\mu_N)$ is:

$$\delta J_2(\mu_N; R - \mu_N) = \lambda_N \left( \mathbb{E}_R[-\ln(1-X)] - \mathbb{E}_{\mu_N}[-\ln(1-X)] \right).$$

Combining the terms, $\delta J = \delta J_1 + \delta J_2$ can be represented in the integral form, as in Theorem C.3, by the following function $g_{\mu_N}(x)$:

$$g_{\mu_N}(x) = -\frac{x^{\lambda_K}}{C(\mu_N)} - \lambda_N \ln(1-x). \tag{8}$$

By the KKT Support Condition (Theorem C.3), the optimal distribution $\mu_N^*$ must be supported on the set of global minima of $g_{\mu_N^*}(x)$.

**Step 3: solving for support and probability mass.**   Let $g := g_{\mu_N^*}$ and $C^* := C(\mu_N^*)$. To analyze the minima of $g$, we now establish a crucial property of $\lambda_K$.

**Lemma C.5.** *In the non-trivial regime ($e^{-\varepsilon_K} + e^{-\varepsilon_N} > 1$), the optimal Lagrange multiplier satisfies $\lambda_K > 1$.*

*Proof.* Consider the derivatives of $g(x)$:

$$g'(x) = -\frac{\lambda_K x^{\lambda_K - 1}}{C^*} + \frac{\lambda_N}{1-x}, \quad g''(x) = -\frac{\lambda_K(\lambda_K - 1)x^{\lambda_K - 2}}{C^*} + \frac{\lambda_N}{(1-x)^2}.$$

Note that $C^* > 0$, otherwise we must have $\mu_N^* = \delta_0$. But if $\mu_N^*$ collapses to a point mass, then we either have $\mu_K^* = \mu_N^*$ (infeasible solution) or $\mathrm{KL}(\mu_K^* \| \mu_N^*) = \infty$. We will try to avoid this and solve for a $\mu_N^*$ with non-singleton support.

Suppose $\lambda_K \leq 1$. If $\lambda_K = 1$, $g''(x) > 0$. If $0 < \lambda_K < 1$, then $\lambda_K - 1 < 0$, so the first term is positive, and $g''(x) > 0$. In both cases, $g(x)$ is strictly convex and has a unique global minimum $x^*$. Thus, $\mu_N^* = \delta_{x^*}$, and hence $\mu_K^* = \delta_{x^*}$ by the previous result on $\frac{d\mu_K^*}{d\mu_N^*}(x)$. This implies the KL divergence objective is zero, contradicting the assumption of the non-trivial regime. $\square$

Knowing $\lambda_K > 1$, we analyze the shape of $g(x)$. Since $\lambda_K - 1 > 0$, $x^{\lambda_K - 1} \to 0$ as $x \to 0$, so $g'(0) = \lambda_N > 0$. It follows that $g$ is increasing both as $x \to 0$ and $x \to 1$. We will show that there is at most one local minimum at some $x^* \in (0, 1)$, so $0$ and $x^*$ must both be global minima, with $g(x^*) = g(0) = 0$.

Consider stationary condition $g'(x) = -\frac{\lambda_K x^{\lambda_K - 1}}{C^*} + \frac{\lambda_N}{1-x} = 0$. Note that the first term is negative, the second term is positive, and $g'(x) > 0$ iff $\frac{\lambda_N}{1-x} > \frac{\lambda_K x^{\lambda_K - 1}}{C^*}$, which is true iff $C^* \frac{\lambda_N}{\lambda_K} > x^{\lambda_K - 1}(1 - x) =: h(x)$. Clearly $h(0) = h(1) = 0$, corresponding to $g'(0)$ being positive at these endpoints. Differentiating $h$, we have:

$$h'(x) = x^{\lambda_K - 2}(\lambda_K - 1 - \lambda_K x).$$

Since $x^{\lambda_K - 2}$ is always positive on $(0, 1)$, this derivative only changes sign once from positive to negative, and therefore the equation $C^* \frac{\lambda_N}{\lambda_K} = h(x)$ has at most two solutions in $(0, 1)$, in which case $h(x)$ will start from zero, increase to a local maximum, and then drop back to $0$ at $x = 1$. Hence, there are at most two points where $g'(x) = 0$, the first one being a local maximum and the second a local minimum, which we call $x^*$.

In fact, if the optimal KL divergence were to be finite, the global minimum of $g = 0$ must be attained at both $x = 0$ and $x^*$:

1. If the minimum is uniquely at $x = 0$. Then $\mu_N^* = \delta_0$. This implies $C^* = 0^{\lambda_K} = 0$ (since $\lambda_K > 1$). This leads to $J(\mu_N^*) = \infty$, which is not optimal.

2. If the minimum is uniquely at $x^*$. Then $\mu_N^* = \delta_{x^*}$. This implies $\mu_K^* = \mu_N^*$ and KL $= 0$, a contradiction.

We conclude that $\mu_N^*$ is a two-point distribution:

$$\mu_N^* = (1 - q^*)\delta_0 + q^* \delta_{x^*}.$$

We can now determine $\mu_K^*$ using the relative density in (7): because $\frac{d\mu_K^*}{d\mu_N^*}(0) = 0^{\lambda_K} = 0$, it follows that it is a point mass $\mu_K^* = \delta_{x^*}$.

Now we use the tight constraints to determine $x^*$ and $q^*$ for $\mu_N^*$.

1. Constraint on $\mu_K^*$: $\mathbb{E}_{\mu_K^*}[-\ln X] = \varepsilon_K \implies -\ln(x^*) = \varepsilon_K \implies x^* = e^{-\varepsilon_K}$.

2. Constraint on $\mu_N^*$: $\mathbb{E}_{\mu_N^*}[-\ln(1 - X)] = \varepsilon_N$.

$$(1 - q^*)(-\ln 1) + q^*(-\ln(1 - x^*)) = \varepsilon_N \implies q^* = \frac{\varepsilon_N}{-\ln(1 - x^*)}.$$

The condition $e^{-\varepsilon_K} + e^{-\varepsilon_N} > 1$ ensures that $\varepsilon_N < -\ln(1 - e^{-\varepsilon_K})$, so $0 < q^* < 1$.

**Step 4: verifying KKT conditions.** Note that we calculate $C^* = \mathbb{E}_{\mu_N^*}[X^{\lambda_K}] = q^*(x^*)^{\lambda_K}$ (since $\lambda_K > 1$).

Condition 1: $g(x^*) = 0$.

$$g(x^*) = -\frac{(x^*)^{\lambda_K}}{C^*} - \lambda_N \ln(1 - x^*) = -\frac{1}{q^*} - \lambda_N \ln(1 - x^*) = 0.$$

This yields $\lambda_N = \frac{-1}{q^* \ln(1 - x^*)}$. Since $q^* > 0$ and $\ln(1 - x^*) < 0$, we have $\lambda_N > 0$.

Condition 2: $g'(x^*) = 0$.

$$g'(x^*) = -\frac{\lambda_K (x^*)^{\lambda_K - 1}}{C^*} + \frac{\lambda_N}{1 - x^*} = 0.$$

Rearranging and substituting $C^*$:

$$\lambda_K = \frac{\lambda_N C^*}{(1 - x^*)(x^*)^{\lambda_K - 1}} = \frac{\lambda_N q^*(x^*)^{\lambda_K}}{(1 - x^*)(x^*)^{\lambda_K - 1}} = \frac{\lambda_N q^* x^*}{1 - x^*}.$$

Substituting the expression for $\lambda_N$:

$$\lambda_K = \left( \frac{-1}{q^* \ln(1 - x^*)} \right) \frac{q^* x^*}{1 - x^*} = \frac{-x^*}{(1 - x^*) \ln(1 - x^*)}.$$

We verify that $\lambda_K > 1$. Let $y = 1 - x^* \in (0, 1)$. We need to show $\frac{-(1-y)}{y \ln y} > 1$. Since $y \ln y < 0$, this is equivalent to $-(1-y) < y \ln y$, or $1 - 1/y < \ln y$. Consider the function $h(y) = \ln y - (1 - 1/y)$. Its derivative is $h'(y) = 1/y - 1/y^2 = (y - 1)/y^2$. On $(0, 1)$, $h'(y) < 0$. Since $\lim_{y \to 1^-} h(y) = 0$, we have $h(y) > 0$ for $y \in (0, 1)$. Thus, $\lambda_K > 1$.

Since $\lambda_K > 1$, the shape analysis in Step 3 holds: $g$ has exactly one local maximum and one local minimum in $(0, 1)$, confirming that $\{0, x^*\}$ are indeed the global minima of $g(x)$. The KKT conditions are fully satisfied, and therefore $\mu_K^*$ and $\mu_N^*$ are the global optimum.

## D. Proofs for Section 4.2

### D.1. Binary-output reduction via data processing

We now show that, under the linear error metrics $d^K(\hat{x}) = 1 - \hat{x}$ and $d^N(\hat{x}) = \hat{x}$, we can restrict to binary outputs without loss of optimality.

**Lemma D.1** (Binary-output reduction via data processing). *Let $(\mu_K, \mu_N)$ be any feasible pair of output distributions supported on $[0, 1]$. Define the Markov kernel $T$ from $[0, 1]$ to $\{0, 1\}$ by*

$$T(1 \mid t) = t, \qquad T(0 \mid t) = 1 - t.$$

*Let $\bar{\mu}_K := \mu_K T$ and $\bar{\mu}_N := \mu_N T$ be the induced distributions on $\{0, 1\}$. Then:*

1. *(Constraints preserved) $\mathbb{E}_{\bar{\mu}_K}[1 - \hat{X}] = \mathbb{E}_{\mu_K}[1 - \hat{X}]$ and $\mathbb{E}_{\bar{\mu}_N}[\hat{X}] = \mathbb{E}_{\mu_N}[\hat{X}]$.*

2. *(Objective non-increasing) For every $p \in (0, 1)$,*

$$F_p(\bar{\mu}_K, \bar{\mu}_N) \leq F_p(\mu_K, \mu_N).$$

*Consequently, in the definition of $R_p(\varepsilon_K, \varepsilon_N)$ we may restrict to $\hat{x} \in \{0, 1\}$, and the optimal output distributions are Bernoulli.*

*Proof sketch.* The first part follows from linearity: under $T$, the binary output $Y \in \{0, 1\}$ satisfies $\mathbb{E}[Y \mid \hat{X} = t] = t$, hence $\mathbb{E}[Y] = \mathbb{E}[\hat{X}]$ and $\mathbb{E}[1 - Y] = \mathbb{E}[1 - \hat{X}]$. For the second part, note that the kernel commutes with mixtures: $\bar{\mu}_p = (p\mu_K + (1-p)\mu_N)T = p\bar{\mu}_K + (1-p)\bar{\mu}_N$. By the data-processing (contraction) inequality for KL under a Markov kernel,

$$\mathrm{KL}(\mu_K T \| \mu_p T) \leq \mathrm{KL}(\mu_K \| \mu_p), \qquad \mathrm{KL}(\mu_N T \| \mu_p T) \leq \mathrm{KL}(\mu_N \| \mu_p),$$

and plugging into the definition of $F_p$ gives the claim. $\square$

### D.2. Solving the convex optimization

By Theorem D.1, it suffices to consider binary outputs $\hat{X} \in \{0, 1\}$. Again, we assume the KL divergence is base-$e$ for convenience of analysis; the shape of optimal solutions is unaffected by this choice of base. Hence $\mu_K$ and $\mu_N$ are Bernoulli distributions, which we parameterize by

$$\mu_K = \mathrm{Bern}(a), \qquad \mu_N = \mathrm{Bern}(b),$$

where $a = \Pr(\hat{X} = 1 \mid X \in \mathcal{K})$ and $b = \Pr(\hat{X} = 1 \mid X \notin \mathcal{K})$. This reduction also allows us to apply Theorem 3.3, which provides the desired first-order expansion of the memory lower bound w.r.t. $p$, around $p = 0$. It now suffices to show that the optimal distribution is given by $a = 1 - \varepsilon_K$ and $b = \varepsilon_N$.

The KL divergence minimization problem for Bernoulli distributions reduces to the two-variable convex program

$$\min_{a,b \in [0,1]} \mathrm{KL}\big(\mathrm{Bern}(a) \,\|\, \mathrm{Bern}(b)\big) \quad \text{s.t.} \quad a \geq 1 - \varepsilon_K, \ b \leq \varepsilon_N. \tag{9}$$

We now show that the KL divergence objective is strictly increasing in $a$ and strictly decreasing in $b$ over the feasible region. Recall the closed form

$$\mathrm{KL}\big(\mathrm{Bern}(a) \,\|\, \mathrm{Bern}(b)\big) = a \ln \frac{a}{b} + (1 - a) \ln \frac{1 - a}{1 - b}.$$

For fixed $b \in (0, 1)$, differentiating w.r.t. $a$ gives

$$\frac{\partial}{\partial a} \mathrm{KL}(\mathrm{Bern}(a) \| \mathrm{Bern}(b)) = \ln \frac{a(1 - b)}{(1 - a)b}.$$

In the non-trivial regime $1 - \varepsilon_K > \varepsilon_N$, the feasible set satisfies $a \geq 1 - \varepsilon_K > \varepsilon_N \geq b$, hence $a > b$ throughout the feasible region. Therefore

$$\ln \frac{a(1 - b)}{(1 - a)b} > 0,$$

so the objective is *strictly increasing* in $a$ over the feasible region. Thus the minimum is attained at

$$a^* = 1 - \varepsilon_K.$$

Similarly, for fixed $a \in (0, 1)$, differentiating w.r.t. $b$ gives

$$\frac{\partial}{\partial b} \mathrm{KL}(\mathrm{Bern}(a) \| \mathrm{Bern}(b)) = \frac{b - a}{b(1 - b)}.$$

Since $a > b$ on the feasible region, we have $\partial / \partial b < 0$, so the objective is *strictly decreasing* in $b$ over the feasible region. Hence the minimum is attained at the largest feasible $b$, namely

$$b^* = \varepsilon_N.$$

Combining the two monotonicity statements yields the unique optimizer of (9):

$$\mu_K^* = \mathrm{Bern}(1 - \varepsilon_K), \qquad \mu_N^* = \mathrm{Bern}(\varepsilon_N).$$

### D.3. A hash-based memory-optimal two-sided filter

We now show that the filter lower bound can be achieved by a hash-based construction, which, like Bloom filters and others, is universe-agnostic, in the sense that the memory usage does not depend on the universe size $u$. Our construction is inspired by the optimal one-sided filters introduced in Porat (2009) and Dietzfelbinger & Pagh (2008), based on solving random linear equations over a finite field. We also note that this construction is not efficient in practice.

**Theorem D.2.** *Assume $\varepsilon_N = 1/q$ for some prime power $q$, and fix any $\varepsilon_K \in [0, 1)$. There exists a hash-based two-sided filter scheme (in the random-oracle model) such that as $n \to \infty$, when initialized on any key set $\mathcal{K} \subseteq \mathcal{U}$ with $|\mathcal{K}| = n$, it uses*

$$\mathrm{KL}\big(\mathrm{Bern}(1 - \varepsilon_K) \,\|\, \mathrm{Bern}(\varepsilon_N)\big) + o(1) \quad \textit{bits of space per key}$$

*and, with probability $1 - o(1)$, the resulting filter satisfies $\mathrm{FNR} \leq \varepsilon_K$ on keys and $\mathrm{FPR} = \varepsilon_N$ on every non-key. In particular, the guarantee is uniform over all universe sizes $u \geq n$.*

*Proof.* Let $\mathbb{F}_q$ be the finite field of size $q = 1/\varepsilon_N$. WLOG, let $\mathcal{U} = [u]$. We assume access to a vector-valued random-oracle hash function

$$h : \mathcal{U} \to \mathbb{F}_q^m, \qquad h(i) = (h(i)_1, \ldots, h(i)_m),$$

where $m$ is a parameter to be determined at the end, and the coordinates $h(i)_j$ are i.i.d. uniform over $\mathbb{F}_q$ and independent across distinct $i$'s.

For $y \in \mathbb{F}_q^m$, consider the linear functions defined by $h(i)$'s:

$$\langle h(i), y \rangle := \sum_{j=1}^m h(i)_j \, y_j \ \in \ \mathbb{F}_q.$$

Consider the system of equations in the unknown $y \in \mathbb{F}_q^m$:

$$\langle h(i), y \rangle = 0, \qquad i \in \mathcal{K}, \tag{10}$$

and restrict to nonzero solutions $y \neq 0$.

**Initialization.** Given $(\varepsilon_K, \varepsilon_N)$ and $\mathcal{K}$, the filter calculates the corresponding $m$, sends each $i \in \mathcal{K}$ through the random oracle to obtain $h(i)$'s, and searches for a nonzero vector $y \in \mathbb{F}_q^m$ that satisfies at least $(1 - \varepsilon_K)n$ equations in (10). If such a $y$ exists, the filter stores a binary encoding of $y$. The construction *succeeds* if a vector $y$ meeting this requirement exists.

The memory usage is $m \log q + O(1)$ bits, where $\log$ is base 2.

**Query.** On input $i' \in \mathcal{U}$, the filter gets $h(i') \in \mathbb{F}_q^m$ from the oracle and outputs 1 iff $\langle h(i'), y \rangle = 0$ (and outputs 0 otherwise).

**Error guarantees.** If initialization succeeds, then at least $(1 - \varepsilon_K)n$ keys satisfy $\langle h(i), y \rangle = 0$. Thus for a uniform random key $i' \sim \mathrm{Unif}(\mathcal{K})$ we have

$$\mathbb{P}[\mathsf{Query}(i') = 0] \leq \varepsilon_K.$$

which satisfies the FNR constraint if we convert it into a permutation-invariant membership tester via Remark 2.2.

For false positives, fix any non-key $i' \in \mathcal{U} \setminus \mathcal{K}$. Since $y$ is a function of $\{h(i)\}_{i \in \mathcal{K}}$ only, and $h(i')$ is independent of $\{h(i)\}_{i \in \mathcal{K}}$, we may condition on $y$ and treat it as fixed. The next lemma implies

$$\mathbb{P}[\mathsf{Query}(i') = 1 \mid y] = \frac{1}{q} = \varepsilon_N,$$

hence $\mathrm{FPR} = \varepsilon_N$.

**Lemma D.3.** *For any fixed $y \in \mathbb{F}_q^m$ with $y \neq 0$, if $H_1, \ldots, H_m$ are drawn i.i.d. from $\mathrm{Unif}(\mathbb{F}_q)$, then*

$$\mathbb{P}\left[ \sum_{j=1}^m y_j H_j = 0 \right] = \frac{1}{q}.$$

*Proof.* WLOG, suppose $y_1 \neq 0$. For any $c \in \mathbb{F}_q$,

$$\mathbb{P}\left[ \sum_{j=1}^m y_j H_j = c \right] = \mathbb{P}\left[ H_1 = y_1^{-1}\left( c - \sum_{j=2}^m y_j H_j \right) \right] = \frac{1}{q},$$

since $H_1$ is uniform and independent of $(H_2, \ldots, H_m)$. $\qquad\qquad\square$

**Choice of $m$ and space bound.** Let

$$D := \mathrm{KL}\Big( \mathrm{Bern}(1 - \varepsilon_K) \,\big\|\, \mathrm{Bern}(\varepsilon_N) \Big).$$

Choose any sequence $t_n \to \infty$ with $t_n = o(n)$ (e.g. $t_n = n^{2/3}$), and set

$$m := \left\lceil \frac{nD + t_n}{\log q} \right\rceil. \tag{11}$$

Then the memory is

$$m \log q = nD + t_n + O(1),$$

i.e. $D + o(1)$ bits per key.

It remains to show that initialization succeeds with probability $1 - o(1)$ (hence at least 0.9 for all large $n$).

**Second moment method for success probability.** Let $Y := \mathbb{F}_q^m \setminus \{0\}$, so $|Y| = q^m - 1$. For each $y \in Y$, define

$$N_y := \mathbb{1}\left\{\#\{i \in \mathcal{K} : \langle h(i), y \rangle = 0\} \geq (1 - \varepsilon_K)n\right\}, \qquad Z := \sum_{y \in Y} N_y.$$

Thus $Z \geq 1$ iff there exists a nonzero $y$ satisfying at least $(1 - \varepsilon_K)n$ equations, i.e. initialization succeeds. Let $p_n = \mathbb{P}[N_y = 1]$, then $\mathbb{E}[Z] = (q^m - 1)p_n$.

Fix any $y \neq 0$. For each key $i \in \mathcal{K}$, by Lemma D.3 and independence across $i$, the indicators $\mathbb{1}\{\langle h(i), y \rangle = 0\}$ are i.i.d. $\mathrm{Bern}(\varepsilon_N)$. Therefore, by standard large-deviation bounds, for each $y$,

$$p_n = 2^{-nD + o(n)}. \tag{12}$$

Now we bound the second moment $\mathbb{E}[Z^2]$:

$$\mathbb{E}[Z^2] = \sum_{y \in Y} \mathbb{E}[N_y] + \sum_{\substack{y, y' \in Y \\ y \neq y'}} \mathbb{E}[N_y N_{y'}].$$

Consider the second term for fixed $y, y' \in Y, y \neq y'$. If $y'$ is a nonzero scalar multiple of $y$, then for each $i \in \mathcal{K}$, $\langle h(i), y' \rangle = 0$ iff $\langle h(i), y \rangle = 0$, so $N_{y'} = N_y$ and $\mathbb{E}[N_y N_{y'}] = \mathbb{E}[N_y] = p_n$. Otherwise $y, y'$ are linearly independent, and the mapping

$$h(i) \mapsto \left(\langle h(i), y \rangle, \langle h(i), y' \rangle\right) \in \mathbb{F}_q^2$$

is a linear surjection. Since $h(i)$ is uniform, the events $A_i := \{\langle h(i), y \rangle = 0\}$ and $B_i := \{\langle h(i), y' \rangle = 0\}$ are independent with $\Pr[A_i] = \Pr[B_i] = 1/q$ and $\Pr[A_i \cap B_i] = 1/q^2$. Independence across $i$ then implies that $\{A_i\}_{i=1}^n$ is independent of $\{B_i\}_{i=1}^n$, and thus $X_y$ and $X_{y'}$ are independent: $\mathbb{E}[X_y X_{y'}] = p_n^2$.

Now we count the number of pairs $(y, y')$ of each type. For each $y$ there are exactly $(q - 2)$ distinct nonzero multiples $y' \in \langle y \rangle \setminus \{y\}$, and $|Y| - (q - 1) = q^m - q$ choices of $y'$ not in the one-dimensional subspace $\langle y \rangle$. Therefore,

$$\mathbb{E}[Z^2] \leq (q^m - 1)p_n + (q^m - 1)(q - 2)p_n + (q^m - 1)(q^m - q)p_n^2$$
$$= (q^m - 1)\left((q - 1)p_n + (q^m - q)p_n^2\right).$$

By Paley-Zygmund,

$$\Pr[Z \geq 1] \geq \frac{\mathbb{E}[Z]^2}{\mathbb{E}[Z^2]} \geq \frac{(q^m - 1)p_n}{(q - 1) + (q^m - q)p_n}.$$

This lower bound goes to 1 as $n \to \infty$ since

$$q^m p_n = 2^{m \log q} \cdot 2^{-nD + o(n)} = 2^{t_n + o(n)} \to \infty,$$

and both the numerator and denominator are dominated by this term. $\square$

## E. Experiment Details

We present a more detailed setup of the experiments in Section 4.3.

We optimize the models using AdamW with $\beta_1 = 0$, $\beta_2 = 0.999$, and base learning rate $3 \times 10^{-3}$ with 1000 warm-up steps and cosine decay. The choice of $\beta_1 = 0$ (so the optimizer is similar to RMSprop) is for better optimization performance. Each batch contains all $n$ positive samples (the full set $\mathcal{K}$) and $n$ fresh negatives sampled uniformly from $\mathcal{U} \setminus \mathcal{K}$. The positive and negative samples are evaluated with a custom weight defined by $\lambda_F$ as follows:

$$\mathcal{L} = \frac{\lambda_F}{\lambda_F + 1} \mathop{\mathbb{E}}_{i \sim \mathrm{Unif}(\mathcal{K})} \left[-\ln \hat{x}(i)\right] + \frac{1}{\lambda_F + 1} \mathop{\mathbb{E}}_{i \sim \mathrm{Unif}(\mathcal{U} \setminus \mathcal{K})} \left[-\ln(1 - \hat{x}(i))\right].$$

We set the seed to be 42 for all our experiments, and all models are trained for 30000 epochs, at which point all losses converge to stable values.

*Table 1.* Reproducibility details for the main experiment.

| Category | Setting |
|---|---|
| **Data** | Universe $\mathcal{U}$: alphabet size $|\Sigma| = 26$, length 15, $|\mathcal{U}| = 26^{15}$. |
| | Facts $\mathcal{K}$: $|\mathcal{K}| = 15145$, sampled uniformly without replacement with seed 42. |
| | Negatives: per step sample size 15145, distribution $\mathrm{Unif}(\mathcal{U} \setminus \mathcal{K})$. |
| **Tokenization** | Tokenization: char-level, vocab size $V = 26$. |
| | Positional encoding: sinusoidal (fixed). |
| **Model** | Architecture: Transformer. |
| | 2 Layers, $d_{\mathrm{model}} = 18/24/36$, heads $h = 3/4/6$, FFN dim $d_{\mathrm{ff}} = 72/96/144$. |
| | Activation: GeLU; pre-LayerNorm; no dropout. |
| | Output head: Mean Pooling $\rightarrow$ Layer norm $\rightarrow$ Linear$\rightarrow$Sigmoid producing $\hat{x}(i) \in (0,1)$. |
| | #Trainable parameters: 8767 / 15145 / 33085. |
| | Precision: bf16 |
| **Objective** | Loss: weighted BCE with weights $\lambda_F/(\lambda_F + 1)$, $1/(\lambda_F + 1)$. |
| | Batch composition: all positives + fresh negatives. |
| **Optimization** | Optimizer: AdamW with $(\beta_1, \beta_2) = (0, 0.999)$ and weight decay 0.01. |
| | Learning rate: base LR $3 \times 10^{-3}$; 1000 warm-up steps; cosine decay. |
| | Training length: 30,000 epochs. |
| | Grad clip: 1. |
| | Initialization: Xavier uniform for matrix; PyTorch default for one-dimensional. |
| **Evaluation** | Facts eval: all $n$ facts. |
| | Non-facts eval: sample size $200k$ from $\mathrm{Unif}(\mathcal{U} \setminus \mathcal{K})$. |
| | Histogram/KL is based on the discretized output distribution with 50 bins. |

We consider three different model sizes, corresponding to 0.58 / 1 / 2.18 parameters per fact. For each model size, we train a model with $\lambda_F \in \{0.25, 1, 4, 8\}$ to observe the trade-off between the two types of errors, given a fixed memory budget. In Figure 1, we report the confidence distributions of training the 1 parameter-per-fact model. The other two sizes are reported in Figure 4 and Figure 5, respectively.

We note that across different model sizes and $\lambda_F$, the non-fact distribution consistently exhibits the heavy tail of hallucinations, whenever the fact distribution is sufficiently concentrated.

In Table 1, we report the full experimental setup. Our implementation code will be released publicly.

### E.1. Fine-tuning experiments on synthetic IDs and real ISBNs

To probe whether the rate–distortion trade-off in Theorem 4.1 continues to hold beyond the from-scratch setting of Section 4.3, we run a second family of experiments that LoRA fine-tune (Hu et al., 2022) a pretrained large language model on two membership-testing tasks of contrasting structure: a structure-free random-ID baseline and a real-world ISBN-13 dataset.

**Datasets.** Both tasks use $|\mathcal{K}| = 10000$ facts and a much larger non-fact pool, so the small-$p$ regime of our theory remains in effect. *Synthetic IDs* mirror the regime of Table 1 but use the pretrained model's native tokenizer: each fact is a length-13 string over the decimal digits $\{0, \ldots, 9\}$, so $|\mathcal{U}| = 10^{13}$, and non-facts are sampled uniformly from $\mathcal{U} \setminus \mathcal{K}$. This setting retains the randomness assumption of the theorem and serves as a structure-free baseline. *ISBN* draws $|\mathcal{K}| = 10000$ valid ISBN-13 strings from the Open Library editions dump `ol_dump_editions_2026-02-28.txt`. The extraction script reservoir-samples ISBN entries after removing hyphens and keeping only 13-digit strings that start with `978` or `979`. Non-facts are constructed to match surface statistics while remaining outside $\mathcal{K}$: we sample the first four digits from the observed ISBN prefixes, randomize the next eight digits, recompute the ISBN-13 check digit, and filter out any string in $\mathcal{K}$. This design preserves the structural regularity of real ISBNs (in particular the publisher-prefix substructure a pretrained model may already encode) while still admitting a large, sampleable set of plausible-looking non-facts.

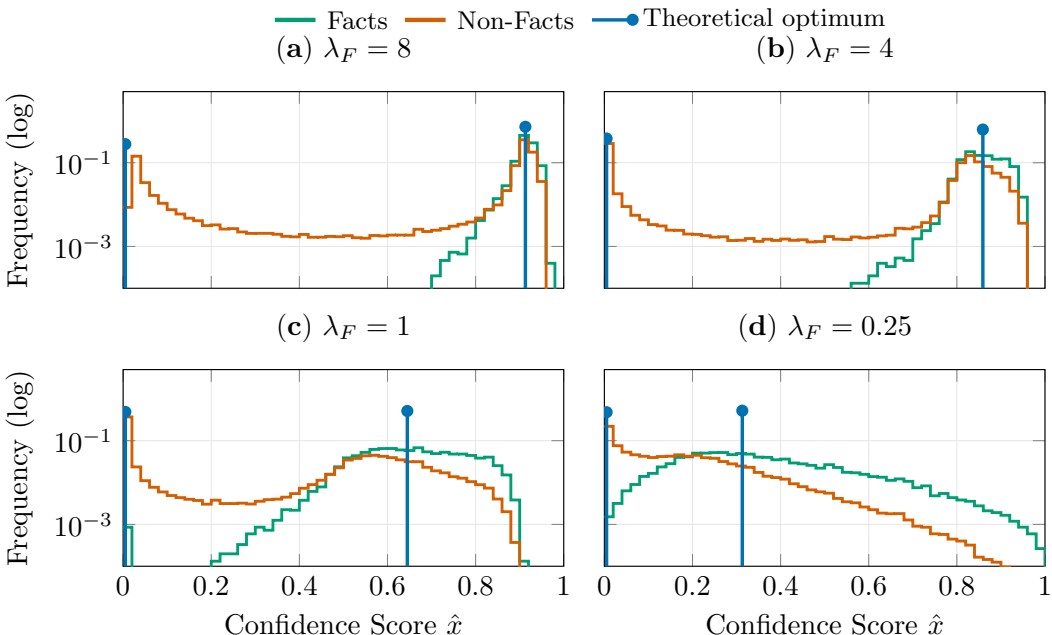

Figure 4. Output distributions with 15145 facts and 8767 parameters.

**Base model and LoRA configuration.** The base model is Qwen/Qwen3.5-2B (Team, 2026), a 2B-parameter pretrained LM with hidden size 2048 and 24 layers arranged as 6 blocks of ($3 \times$ Gated DeltaNet $+ 1 \times$ Gated Attention). We freeze all base weights and inject LoRA adapters on modules matched by q_proj and v_proj, using rank $r \in \{2, 4\}$, scaling $\alpha = 2r$, and dropout 0.05. This adds 104448 trainable parameters at $r = 2$ and 208896 at $r = 4$, i.e., on the order of $10^{-4}$ of the base model's parameter count. We use the same weighted binary cross-entropy objective as the from-scratch experiments and sweep $\lambda_F \in \{0.5, 1, 2, 4\}$ to trace the empirical rate–distortion frontier under each rank. Full hyperparameters are listed in Table 2.

**Empirical results.** Figures 6a to 6c plot representative empirical confidence distributions on facts (green) and non-facts (red) under $\lambda_F \in \{1, 4\}$ for all three configurations. Across all six panels the qualitative pattern predicted by Theorem 4.1 recurs: facts concentrate near a single high-confidence atom close to $x^\star = e^{-\varepsilon_K}$, and a non-negligible mass of non-facts piles up at the *same* high-confidence region rather than spreading toward zero. The shape is less well-aligned with theoretical optimum compared to the from-scratch synthetic training setup, mainly due to the sub-optimality of LoRA fine-tuning for this task.

Quantitatively, the empirical binned KL falls within 14–22% of the information-theoretic lower bound at the observed $(\varepsilon_K, \varepsilon_N)$, with all twelve runs clustering around a slope-$\approx 1.18$ line (Figure 3). The prior-bearing ISBN runs and the prior-free Synthetic runs occupy the same overhead band, suggesting that the pretraining prior does not measurably shift the rate–distortion frontier on this task. Training-loss curves for one representative shared setting, $\lambda_F = 1$ and rank 4 (Figure 7), show the main effect of pretraining: the pretrained model moves rapidly into the low-loss regime, reinforcing that pretraining mainly improves optimization rather than the final rate–distortion frontier.

**Parameter efficiency and connection to the effective memory budget.** Reading Figure 3 in absolute terms, the empirical KL stored per trainable LoRA parameter averages $\approx 0.17$ bits/param at rank 4 and $\approx 0.27$ bits/param at rank 2.[4] Both are an order of magnitude below the $\approx 2$ bits/param attained by the from-scratch models in Section 4.3 (a value itself consistent with Allen-Zhu & Li (2025) on random data). The drop is expected—LoRA expresses only a low-rank update on top of frozen base weights, so it mostly makes local adjustments around the pretrained predictor rather than globally reshaping the model's output distribution for a new random fact table—but it also gives a direct empirical handle on the distinction emphasized in Section A: the operative quantity in Theorem 3.1 is $I(W; \mathcal{K})$, the information the model can actually allocate to $\mathcal{K}$, not $|W|$ or the nominal trainable count. The factor $\sim 10\times$ gap between LoRA and from-scratch makes this distinction quantitative on the same task family.

---

[4]For example, at $r = 4$ on ISBN with $\lambda_F = 1$: empirical KL $\approx 3.44$ bits per fact $\times$ 10000 facts $\div$ 208896 LoRA params $\approx 0.16$ bits/param.

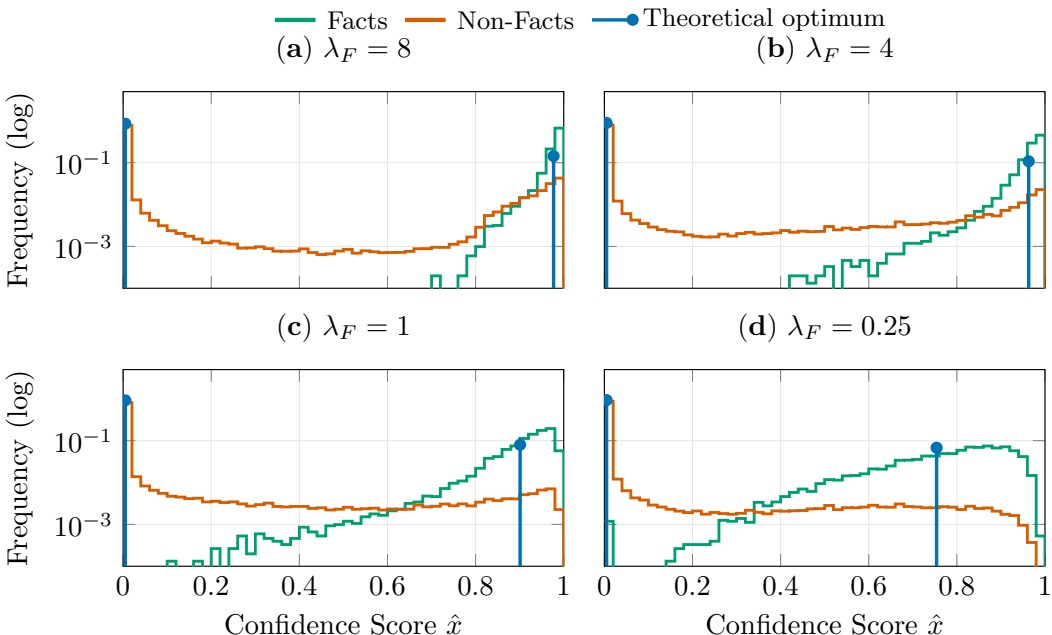

*Figure 5.* Output distributions with 15145 facts and 33085 parameters.

### E.2. Effect of Key-to-Universe Ratio $p$ on Synthetic Experiments

Theorem 3.1 is stated in the sparse limit $p \to 0$, while Theorem 3.3 gives the first-order correction away from this limit; our main experiments fix $n = 15145$ with $p \approx 0.003$. To check robustness at larger densities, we vary $p = n/u \in \{0.003, 0.03, 0.3\}$ by shrinking $u$ while keeping $n$ fixed, under the same 2-layer transformer and weighting scheme as Table 1, with $\lambda_F \in \{1, 4\}$.

The qualitative picture is unchanged across all six runs (Figure 8): facts concentrate near the predicted high-confidence atom and non-facts develop the predicted hallucination tail, with empirical and theoretical optima aligning closely. Table 3 reports the empirical and lower-bound bits per key. The empirical/lower-bound ratio stays close to one for small $p$ and widens modestly at $p = 0.3$; we attribute this widening to changed optimization geometry away from the sparse regime, not to the first-order $O(p)$ correction alone. Separately, the total loss is lower at larger $p$ because the memory task becomes easier when true facts occupy a larger fraction of the universe. This shows that our theoretical predictions remain accurate until true facts become a substantial fraction of potential facts.

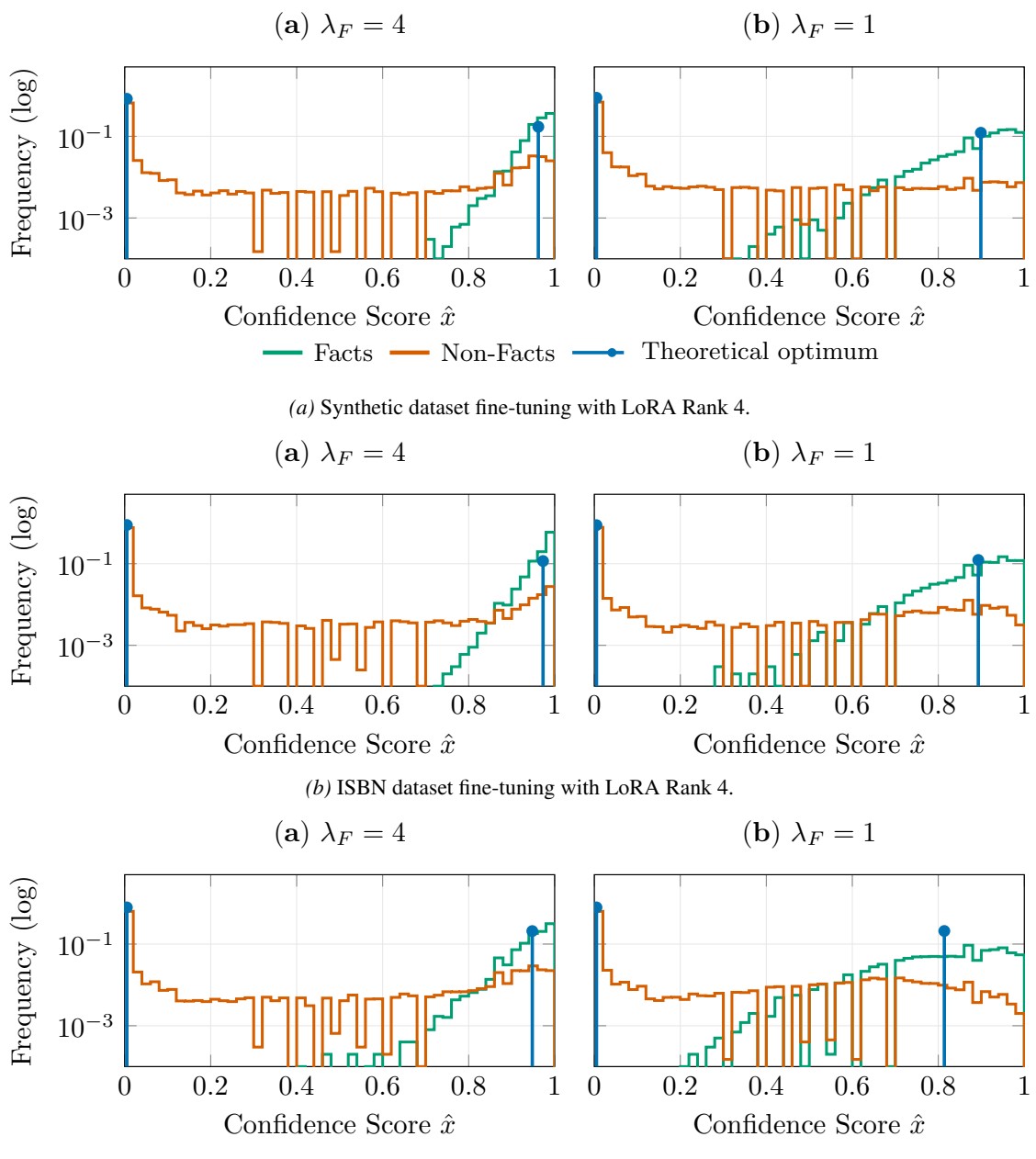

*(a)* Synthetic dataset fine-tuning with LoRA Rank 4.

*(b)* ISBN dataset fine-tuning with LoRA Rank 4.

*(c)* ISBN dataset fine-tuning with LoRA Rank 2.

*Figure 6.* Representative output distributions on facts vs. non-facts across $\lambda_F \in \{1, 4\}$ for the three LoRA fine-tuning configurations.

*Table 2.* Reproducibility details for the LoRA fine-tuning experiments (Section E.1).

| Category | Setting |
|---|---|
| **Data (synthetic)** | Each fact is a length-13 string over decimal digits $\{0, \ldots, 9\}$; $\|\mathcal{U}\| = 10^{13}$. 
 $\|\mathcal{K}\| = 10000$, sampled uniformly without replacement, seed 42. 
 Train non-facts: 10000 fresh samples per epoch from $\mathrm{Unif}(\mathcal{U} \setminus \mathcal{K})$. Eval non-facts: a fixed set of 20000 samples from $\mathcal{U} \setminus \mathcal{K}$. |
| **Data (ISBN)** | Facts: $\|\mathcal{K}\| = 10000$ valid ISBN-13 strings in the fixed file `positive_10k_isbns.txt`, generated by reservoir sampling from `ol_dump_editions_2026-02-28.txt` after filtering to 13 digits and prefixes `978/979`. 
 Train non-facts: 10000 fresh samples per epoch; eval non-facts: a fixed set of 20000 samples. Both are generated by sampling a real four-digit prefix, randomizing the next eight digits, recomputing the ISBN-13 check digit, and filtering to exclude $\mathcal{K}$. |
| **Tokenization** | Native Qwen3.5 BPE tokenizer (vocab size 248,320); IDs are tokenized as plain text with no special preprocessing. |
| **Base model** | `Qwen/Qwen3.5-2B`: 24 layers, hidden size 2048, hybrid layout of 6 blocks each containing 3 Gated DeltaNet layers and 1 Gated Attention layer. Pretrained weights frozen throughout. 
 Precision: bf16. |
| **LoRA adapters** | Rank $r \in \{2, 4\}$; scaling $\alpha = 2r$ (so $\alpha = 4$ at $r = 2$ and $\alpha = 8$ at $r = 4$); dropout 0.05. 
 Target modules: `q_proj`, `v_proj`. 
 Trainable parameters: 104448 at $r = 2$, 208896 at $r = 4$. |
| **Objective** | Weighted BCE: $\mathcal{L} = \frac{\lambda_F}{\lambda_F + 1} \mathbb{E}_{i \sim \mathrm{Unif}(\mathcal{K})} \left[ -\ln \hat{x}(i) \right] + \frac{1}{\lambda_F + 1} \mathbb{E}_{i \sim \mathrm{Unif}(\mathcal{U} \setminus \mathcal{K})} \left[ -\ln(1 - \hat{x}(i)) \right]$. 
 $\lambda_F$ settings: $\{0.5, 1, 2, 4\}$. |
| **Optimization** | Optimizer: AdamW, base LR $3 \times 10^{-4}$, weight decay 0.01, gradient-norm clip 1.0. 
 LR schedule: cosine with warm-up ratio 0.05. 
 Batch size 64, gradient accumulation 1, 80 epochs. Seed 42. |
| **Evaluation** | Facts: full training key set $\|\mathcal{K}\| = 10000$. Non-facts: 20000 disjoint samples. 
 Empirical KL computed on 50-bin histograms of $\hat{x}$ over facts vs. non-facts. |

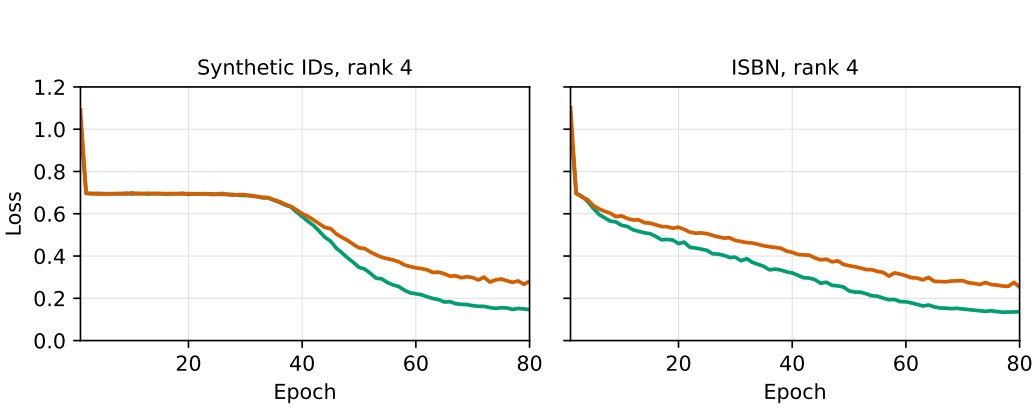

LoRA fine-tuning training curves ($\lambda_F = 1$, rank 4)

*Figure 7.* Training loss curves (facts loss vs. non-facts loss) under the shared setting $\lambda_F = 1$ and LoRA rank 4 for Synthetic IDs and ISBN. Both fine-tuning runs enter the low-loss regime quickly, consistent with pretraining mainly improving optimization speed rather than shifting the final rate–distortion frontier.

*Table 3.* Empirical vs. information-theoretic lower bound on bits per key, across key-to-universe ratios $p$ and fact weights $\lambda_F$. The last column reports the empirical-to-lower-bound ratio; its modest widening at $p = 0.3$ reflects changed optimization geometry away from the sparse regime.

| $p$ | $\lambda_F$ | Empirical (bits) | Lower bound (bits) | Ratio |
|-----|-----|-----|-----|-----|
| 0.003 | 1.0 | 1.810 | 1.612 | 1.12 |
| 0.003 | 4.0 | 1.575 | 1.313 | 1.20 |
| 0.03 | 1.0 | 1.948 | 1.735 | 1.12 |
| 0.03 | 4.0 | 1.655 | 1.413 | 1.17 |
| 0.3 | 1.0 | 3.857 | 2.754 | 1.40 |
| 0.3 | 4.0 | 2.949 | 2.040 | 1.45 |

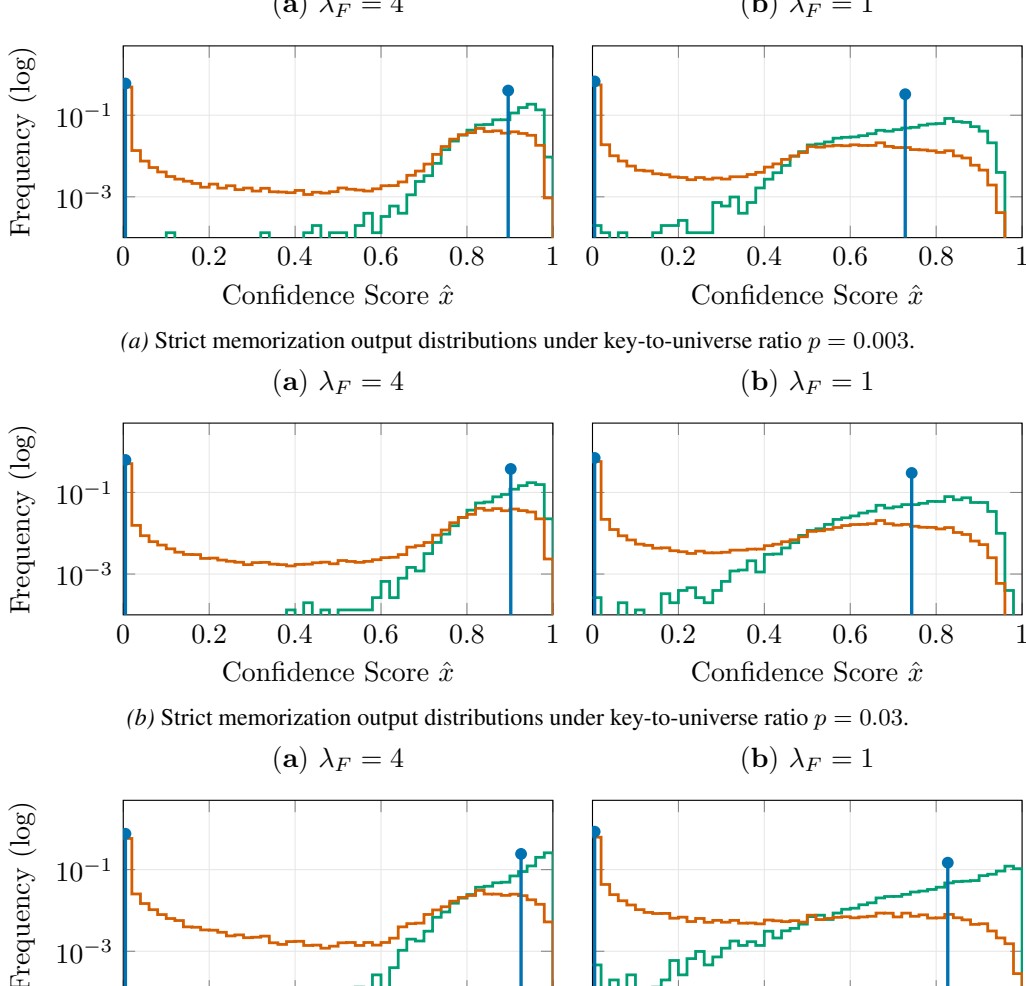

(a) Strict memorization output distributions under key-to-universe ratio $p = 0.003$.

(b) Strict memorization output distributions under key-to-universe ratio $p = 0.03$.

(c) Strict memorization output distributions under key-to-universe ratio $p = 0.3$.

*Figure 8.* Output distributions on facts vs. non-facts across different choice of weight $\lambda_F$ ($\lambda_F = 4.0$ and $\lambda_F = 1.0$) under different key-to-universe ratios $p \in \{0.003, 0.03, 0.3\}$ for the synthetic strict memorization task.

