# OpenReview forum: "Hallucination is a Consequence of Space-Optimality: A Rate-Distortion Theorem for Membership Testing"
_ICML.cc/2026/Conference — ICML 2026 spotlight_

### Official Review · Reviewer_qpNF · 2026-03-03

**Soundness:** 4
**Presentation:** 3
**Significance:** 4
**Originality:** 3
**Overall Recommendation:** 5
**Confidence:** 5

**Summary:**

The paper presents a rigorous theoretical framework for understanding hallucinations in LLMs through the lens of information theory, modeling LLMs as "membership testers" with finite memory capacity. By formalizing factuality judgment as a membership testing problem in a sparse, closed-world regime, the authors establish a rate-distortion theorem showing that the optimal memory efficiency per key is characterized by the minimum KL divergence between the score distributions of facts and non-facts. A critical finding is that the information-theoretically optimal strategy under limited capacity is to assign high confidence to a fraction of non-facts, making them indistinguishable from facts. This framework proves that a fundamental Pareto frontier exists between forgetting (false negatives) and hallucinating (false positives). Thus, any attempt to eliminate hallucinations simply shifts the model along this frontier, requiring an increase in either memory budget or in the refusal rate. The authors illustrate this theory with synthetic experiments, showing how varying the importance of fact-loss versus non-fact-loss causes the model to traverse this memory-error trade-off.

**Compliance With Llm Reviewing Policy:**

Affirmed.

**Final Justification:**

This paper provides a rigorous information-theoretic framework that successfully characterizes LLM hallucinations as an unavoidable consequence of finite memory capacity. The establishment of a fundamental Pareto frontier between forgetting and hallucinating offers a profound, well-supported theoretical contribution to the field. Given the originality of the formulation and the soundness of the mathematical derivations, I recommend the paper for acceptance.

**Key Questions For Authors:**

1. The authors suggest that RAG mitigates hallucinations because the memory budget is no longer a limiting factor when non-parametric memory is present. However, if RAG is viewed as an extension of membership testing with a larger but still finite capacity, does it fundamentally move the Pareto frontier, or does it simply shift the model to a different point on the same memory-error trade-off curve?
2. In the context of practical LLM training, what is the real-world equivalent of the $\lambda_F$ weighting parameter used in the synthetic experiments? Furthermore, can this theoretical understanding of weight-tuning for fact versus non-fact loss be directly applied to loss function design to produce models at specific target points along the trade-off curve?
3. While the paper focuses on "random facts" that are unstructured and non-generalizable, is this classification-based view too restrictive? How does the existence of structured knowledge, such as reasoning or linguistic syntax, affect the applicability of the rate-distortion theorem in real-world scenarios?
4. The training objective incorporates a specific loss for non-facts to penalize hallucinations. Can this be theoretically linked to DPO? Do such methods that utilize positive and negative pairs simply shift a model along the trade-off curve?
5. Often, elements are categorized into true claims, false claims, and non-claims (irrelevant or nonsensical input). Does the theory hold under such a view where false claims also occupy a sparse or near-zero measure in the universe of plausible statements?

**Limitations:**

1. The paper does not include a dedicated section on limitation.

2. The authors rely on a permutation invariance assumption but they provide little discussion on how to relax this assumption for practical, non-invariant LLMs.

3. The empirical validation is limited to synthetic experiments on random character strings, leaving a gap in evidence regarding how the theory translates to real-world tasks or fine-tuning on high-dimensional natural language datasets.

4. The work establishes profound theoretical conclusions, such as the inevitability of the hallucination channel, but offers insufficient guidance for practitioners on how to navigate these trade-offs in real-world model development or deployment.

**Strengths And Weaknesses:**

Strengths
1. The paper introduces a highly original formulation by modeling LLM factuality as a membership testing problem. This provides a robust information-theoretic framing that allows for a precise characterization of hallucination.
2. The presentation is exceptionally detailed, featuring sound mathematical derivations and a formal rate-distortion theorem that characterizes memory-error trade-offs.
3. The work offers a profound conclusion: under finite capacity, the optimal strategy for a model is to assign high confidence to certain non-facts, making them indistinguishable from true facts. This suggests that hallucinations are an inherent consequence of lossy compression and cannot be entirely eliminated.
4. The synthetic experiments provide a clear visualization of the theoretical findings. Specifically, the Pareto frontier illustrated in the weight sweep ($\lambda_F$) effectively demonstrates how models traverse the trade-off between forgetting and hallucinating

Weaknesses
1. While the proofs are rigorous, the paper lacks sufficient high-level intuition and visual aids to guide the reader through the complex derivations.
2. The impact of the paper would be strengthened by reordering the content. Starting with the synthetic experiments and the memory-error frontier (Figure 2a) would provide immediate context before diving into abstract theorems. Additionally, the section on the "hallucination-free" regime contains some of the paper's most critical insights but feels under-emphasized.
3. The frequent use of small subsections disrupts the narrative flow and makes it difficult to follow the primary argument. Instead of building a story, it creates chunks of information that hinder the reading experience.
4. The evaluation is limited to synthetic data. The paper would benefit from experiments involving standard LLMs fine-tuned on factual tasks to demonstrate how the theory applies to non-synthetic, high-dimensional language data.

Minor Issues
1. Figure 1 is not self-contained and its significance (showing the overlap of fact and non-fact mass) is only fully understandable after reading the experimental results in Section 4.3.
2. Figure 2a represents the core operational message of the paper and should likely appear earlier to anchor the theoretical discussion.
3. The relationship between the sequence of membership testers $\{M_j\}$ and the general tester $M$ is not explicitly defined upon first mention.
4. Key parameters, such as $p$ and $\lambda$, are utilized in theorem statements before being formally defined in the text.

---

> ### Author Rebuttal · Authors · 2026-03-31
>
> We thank the reviewer for the thorough review and constructive suggestions. We address the questions and weaknesses below.
>
> ---
>
> ***Presentation and limitations.***
>
> We appreciate the detailed suggestions on readability. In our revision, we will merge the minor `\paragraph{}` subsections to improve narrative flow, reorder content to front-load Figure 2a, and better emphasize the "hallucination-free" regime. We will also add a dedicated Limitations section describing the scope and key assumptions, including how to connect the permutation-invariance assumption to real-world LLMs.
>
> ---
>
> ***Additional experiments for bridging the theory to LLM.***
>
> We agree that bridging toward more realistic settings strengthens the paper. We are currently extending the experiments in two directions: (1) LoRA fine-tuning of a pretrained LLM to recognize synthetic random ID numbers, where the LoRA rank can serve as controllable proxies for the memory budget; and (2) LoRA fine-tuning on a random subset of real-world ISBN numbers, which tests how pretrained priors (e.g., learned publisher-prefix structure) interact with the memorization task.
> Preliminary ISBN output distributions are qualitatively consistent with the predicted tradeoff, with roughly 20% KL divergence overhead relative to the corresponding theoretical optimum. Interestingly, LoRA fine-tuning exhibits a substantially lower parameter efficiency, storing only ~0.22 bits/param of information compared to the ~2 bits/param of our from-scratch models. This also supports our argument that the effective memory budget for random facts is far smaller than the raw parameter count.
>
> ---
>
> ***Why prioritize controlled tasks over QA benchmarks***
>
> We chose ISBNs over evaluating production models on factual QA benchmarks (like TriviaQA) to better isolate the mechanism described by our theory. Our theory applies to facts that are random and non-inferable, meaning they cannot be deduced from other knowledge. Standard QA benchmarks violate this assumption: questions exhibit dependencies (knowing one answer helps with another), many facts are partially inferable from broader world knowledge, and the effective memory budget allocated to these benchmark questions within a production model is neither measurable nor controllable. Such experiments would not be well-posed to test our theory.
>
> ---
>
> ***Answers to key questions***
>
> **Q1 (RAG and the Pareto frontier):** RAG fundamentally shifts the Pareto frontier by introducing external memory, effectively increasing the total budget. When external data is informative, this moves the frontier to a much more favorable position. We agree with your assessment that, since the combined system remains finite, errors still persist, just at a reduced level.
>
> **Q2 ($\lambda_F$ and loss design):** $\lambda_F$ corresponds most naturally to supervised fine-tuning, where one reweights factual QA pairs against refusal examples (teaching the model to say "I don't know"). Our synthetic results suggest that reweighting fact vs. non-fact supervision can traverse the tradeoff, though further validations remain open for future work. If true, this could imply practical loss function designs: For example, practitioners can increase weight on negative examples to further control hallucinations in high-stakes applications.
>
> **Q3 (Structured vs. unstructured knowledge):** The classification view is a deliberate simplification to reveal fundamental limits, similar to Kalai et al. For structured knowledge, we expect the specific tradeoff expression and the optimal shape to change, but factual knowledge always leaves an incompressible residual (e.g., specific names or dates). Structure can reduce the effective count of independent facts and potentially break the symmetry among the facts, but it does not eliminate the tradeoff and our core observation: when potential facts vastly outnumber true facts, hallucinations remain hard to eliminate and the more preferable mode of error.
>
> **Q4 (Link to DPO):** With similar training hyperparameters, we posit that the memory budget on a particular class of random facts should be similar. Using asymmetric DPO with different weights would therefore indeed shift a model along the existing tradeoff curve. It would be an interesting future direction to study this connection.
>
> **Q5 (False claims vs. non-claims):** Our framework accommodates this view: our non-facts are defined specifically as plausible claims that "look like" facts, in the absence of relevant memorization. For instance, a model with linguistic knowledge may easily reject gibberish (non-claims) but still accept incorrect yet plausible biographies. Because there are vastly more plausible-but-false claims than true facts, our rate-distortion theorem fully applies to the challenge of distinguishing truth from plausible falsehoods.

---

> > ### Author Rebuttal · Reviewer_qpNF · 2026-04-03
> >
> > I appreciate the detailed rebuttal provided by the authors. The paper offers a significant contribution to the field, and I maintain my recommendation for acceptance.

---

### Official Review · Reviewer_pZSw · 2026-03-09

**Soundness:** 3
**Presentation:** 2
**Significance:** 3
**Originality:** 3
**Overall Recommendation:** 4
**Confidence:** 3

**Summary:**

The paper studies why LLMs may produce high-confidence hallucinations on random facts. The authors set up the problem as a sparse membership-testing problem over a large universe of possible facts. The main theoretical result is a rate-distortion characterization showing that, in the sparse limit, the optimal memory cost depends on the KL divergence between score distributions for true facts and non-facts. Under log-loss, the authors show the optimal solution gives a population of non-facts the same high-confidence score as true facts. This result extends to binary decisions. Experiments on small transformers trained on synthetic random strings qualitatively match the formal guarantees.

**Compliance With Llm Reviewing Policy:**

Affirmed.

**Final Justification:**

I maintain my recommendation for weak acceptance, as the complex experimental testbed (involving LoRA) is still ongoing, and the current preliminary results only partially address my second concern.

**Key Questions For Authors:**

- For a fixed-weight model, is the main takeaway that post-processing alone cannot reduce hallucinations and that meaningful gains require adding external memory or retrieval?
- Is there any work that links the theoretical results with the assumption that pretrained LLM weights encode only limited information about random, non-inferable facts?

**Limitations:**

The paper implicitly acknowledges that its theory concerns random facts in a closed-world setting. However, a dedicated Limitations section would strenghten the paper's presentation.

**Strengths And Weaknesses:**

*Strengths*

- The paper provides a new theoretical result: the authors show for sparse random facts under limited effective memory, hallucination is an information-theoretically optimal distortion pattern: with log-loss training, the model gives some non-facts high confidence, and with binary decisions, thresholding only moves along the usual false-positive/false-negative frontier.
- The authors extend the theoretical findings to an empirical setup. The learned score distributions exhibit the formally derived heavy non-fact tail overlapping the fact mass, and the binned empirical KL corroborate the information-theoretic lower bound.

*Weaknesses*
- The main result is proved on the assumption of sparse random facts in a closed-world membership-testing abstraction, and the key assumption that real LLMs have a similarly small effective memory budget for such facts is argued heuristically in the appendix
- The empirical validation is limited and slightly under-documented. It uses a single synthetic setup with 2-layer transformers. More complex settings and models would strengthen the theoretical results.

---

> ### Author Rebuttal · Authors · 2026-03-31
>
> We thank the reviewer for the careful reading and constructive feedback. We address the two main concerns below, and respond to the key questions.
>
> ---
>
> ***Limitation section and writing***
>
> We will add a limitation section that outlines the key assumptions and bridges that are needed to connect our theory to reality. We will also improve the readability (e.g. heavy notation), as suggested by most reviewers.
>
> ---
>
> ***On the scope and purpose of the memory-budget assumption***
>
> We would like to clarify that the validity of our theorems does not depend on any assumption about the magnitude of the memory budget. Theorem 3.1 is an unconditional lower bound: for *any* finite memory budget $\mathcal{M}$ and a sparse, large universe of potential facts, the KL-divergence-based rate-distortion bound holds in the limit, and both the shape of the optimal strategy (section 4.1) and the Pareto frontier between hallucination and forgetting (section 4.2) are natural consequences of the *sparsity* and *randomness* of facts. The main takeaway from this idealized model is that hallucination is the optimal and even necessary mode of error for a capacity-constrained model.
>
> The "small effective memory budget" argument in the appendix serves a different purpose: it motivates *regime relevance*, i.e., that real LLMs are unlikely to have allocated sufficient capacity to random, non-inferable facts to escape the constrained regime. This is indeed a heuristic bridge, which we acknowledge. But our central theoretical contribution serves to establish fundamental limits without requiring that all practical systems operate near those limits, similar to other theoretical works that attempt to explore the fundamental drivers of hallucination.
> That said, we agree it is valuable to provide more concrete empirical grounding for the regime-relevance claim, and we address this below.
>
> ---
>
> ***New experiments: LoRA fine-tuning on real-world identifiers***
>
> We have started extending the empirical evaluation along exactly the directions raised by the reviewer. In the revised version, we will include: (1) LoRA fine-tuning of a pretrained LLM on synthetic random ID membership data, where LoRA rank and $\lambda_F$​ are swept to serve as controllable proxies for the memory budget and fact/non-fact tradeoff respectively; and (2) LoRA fine-tuning on a random subset of real-world ISBN numbers, which introduces some structured prior knowledge while still retaining a substantial random-ID component that is closer to the regime our theory is intended to probe.
>
> Interestingly, our preliminary LoRA results also provide additional evidence in answering Q2 (why limited memory for random facts). Specifically, fine-tuning exhibits a substantially lower parameter efficiency, storing only ~0.22 bits/param of information compared to the ~2 bits/param of our from-scratch models.
>
> ---
>
> ***Answers to key questions***
>
> **Q1 (Post-processing and hallucination reduction):**
>
>  Yes, this is the correct interpretation. For a fixed model, post-processing (e.g., thresholding, calibration) moves the model along the existing Pareto frontier -- trading hallucinations for refusals -- but cannot expand it. Meaningful reduction of the total error rate requires either increasing the memory budget (more parameters, more specialized training, RAG) or reducing the effective number of facts to memorize.
>
> **Q2 (Empirical link between theory and limited LLM memory for random facts):**
>
> The most direct evidence comes from "Understanding LLM Behaviors via Compression" [Pan et al.], which models syntax and knowledge as competing for a finite memory budget measured in mutual information. In their experiments, 400,000 synthetic profiles are generated with 50 templates per attribute type; even the largest model (~253M parameters) achieves only ~40% accuracy on entities appearing just 4 times in training, while a 7.2M model hallucinates almost all entities. Notably, below each model's frequency threshold, the model would hallucinate by producing grammatically correct but factually wrong profiles, consistent with the prediction of our framework that hallucination is the natural error mode under capacity constraints. Their theoretical predictions, based on mutual-information-measured memory budgets, closely match these empirical findings. This is consistent with our regime-relevance claim: for long-tailed, random-looking facts (e.g., specific legal cases, bibliographical entries) where hallucinations are prevalent, the effective memory budget measured by mutual information is indeed very small. The cited works [Huang et al. 2025] and [Arpit et al. 2017] also provide complementary evidence that random and rare samples are memorized last, though using different memorization measures not directly based on mutual information.

---

> > ### Author Rebuttal · Reviewer_pZSw · 2026-04-03
> >
> > Thank you for the detailed rebuttal. I maintain my recommendation for weak acceptance, as the complex experimental testbed (involving LoRA) is still ongoing, and the current preliminary results only partially address my second concern.

---

### Official Review · Reviewer_33hH · 2026-03-12

**Soundness:** 3
**Presentation:** 2
**Significance:** 3
**Originality:** 3
**Overall Recommendation:** 5
**Confidence:** 3

**Summary:**

This paper studies hallucination in large language models from an information-theoretic perspective. The authors model factual knowledge as a membership testing problem, where the model must determine whether a candidate statement belongs to a set of known facts. In the sparse regime where facts are rare within a large universe of possible statements, the paper derives a rate–distortion theorem showing that the minimum memory required per stored fact equals the minimum KL divergence between the score distributions assigned to facts and non-facts. The analysis leads to a key insight: under limited memory capacity, the optimal strategy for minimizing log-loss assigns high confidence to all true facts but also to a non-zero fraction of non-facts, producing a “hallucination channel.” The paper studies two operational settings, probability estimation with log-loss and binary decision via thresholding, and shows that both obey the same memory–error trade-off. Synthetic experiments with random string facts support the theory and show that limited model capacity can naturally lead to confident hallucinations.

**Compliance With Llm Reviewing Policy:**

Affirmed.

**Final Justification:**

The paper provides a technically sound and original perspective by linking hallucination to approximate membership testing and rate–distortion theory, showing it can arise as an optimal error under limited memory. The main weaknesses are limited empirical validation and heavy formalism.

The rebuttal addresses these concerns through planned ISBN-based experiments, finite-(p) analysis, and added intuition for each theorem. This improves clarity and strengthens the empirical direction.

Overall, I increased my score to accept and encourage including these additions in the final version.

**Key Questions For Authors:**

Please refer to the weaknesses section for the main points that would benefit from further clarification by the authors.

**Limitations:**

yes

**Strengths And Weaknesses:**

### **Strengths**

- Provides a unified theoretical framework connecting LLM hallucination on random facts with approximate membership testing and rate–distortion theory, yielding a clear result that the per-key memory cost equals a constrained KL divergence in the sparse regime.

- Identifies a concrete *“hallucination channel”* under log-loss, explaining why high-confidence false positives can arise as a capacity-optimal error mode even with perfect training data and optimization.

- Establishes connections to classical space lower bounds for approximate membership filters, clarifying constants, achievability, and how these results relate to LLM decision rules based on score thresholding.

- Uses a synthetic setup with a very large universe and random keys that approximates the sparse asymptotic regime, showing qualitative and quantitative agreement with the theory (about 12% KL overhead).





### **Weaknesses**

- The paper is difficult to read due to its heavy formalism and limited intuitive explanations. Additional intuition, examples, and clearer exposition would make the theoretical results more accessible to a broader ML audience.

- Empirical validation is limited to a synthetic random-string setting with small Transformer models. The paper does not evaluate the predictions in more realistic scenarios such as factual QA benchmarks, calibration settings, or abstention-based systems where the proposed thresholding and memory–error trade-offs could be tested operationally.

- The experiments operate strictly in the extremely sparse regime ( $n/u \ll 1$) and do not explore finite-\(p\) effects, despite Theorem 3.3 providing a characterization of such corrections. Sensitivity to universe size \(u\), number of facts \(n\), and model architecture is therefore largely unexplored.

- The theoretical framework assumes a simplified random-facts regime where knowledge is non-generalizable and uniformly distributed. It remains unclear how well this assumption captures real-world factual knowledge in language models, which often exhibits structure and redundancy.

---

> ### Author Rebuttal · Authors · 2026-03-31
>
> We thank the reviewer for recognizing the value of our unified theoretical framework and for the constructive feedback on our presentation. We address the weaknesses and questions below.
>
> ---
>
> ***Heavy formalism, limited intuition***
>
> We agree that the heavy formalism poses an accessibility barrier. In the revised version, we will add dedicated intuition paragraphs after each main theorem, explaining the key quantities and results in plain language. For instance, after Theorem 3.1 we will include a paragraph explaining how the rate-distortion result reduces space lower bounding to the problem of finding optimal $\mu_K, \mu_N$, and how this relates to the filter lower bound.
>
> ---
>
> ***On the role of the synthetic experiments***
>
> Our paper establishes model-agnostic results: that hallucination is the optimal form of error (Section 4.1) and is in fact inevitable (Section 4.2) when the universe of potential facts is large and memory is limited. Our current synthetic setup was chosen to isolate the regime covered by the theory as cleanly as possible. It serves to demonstrate that the predicted three-way tradeoff between hallucination, forgetting, and memory is not a purely mathematical artifact, but is realized by a simple transformer trained with standard methods.
>
> That said, we agree that bridging towards more realistic settings strengthens our paper, as further discussed in the next section below. However, the empirical bridge needs to be chosen carefully to match the regime of the theory. For example, while evaluation on standard QA benchmarks would be useful for a broader context, it is not by itself a clean test of our theory. Benchmark answers exhibit inter-question dependencies and are partially inferable from world knowledge, violating our randomness assumption. More critically, neither the effective fact count nor the per-fact memory budget is measurable or controllable in a production LLM, which makes it difficult to map an observed accuracy curve cleanly onto the predicted tradeoff.
>
> ---
>
> ***Further experiments and preliminary results***
>
> Many categories of real-world knowledge (e.g. legal case citations, bibliographic metadata, or identifier codes) carry substantial residual randomness even after accounting for structural regularities, placing them closer to the regime our theory describes. We are currently extending the experiments in two directions: (1) LoRA fine-tuning of a pretrained LLM on synthetic random ID membership data, where the LoRA rank serves as a controllable proxy for memory budget; and (2) LoRA fine-tuning on a subset of real-world ISBN numbers, where ground truth is exactly known, $\mathcal{K}$ is controlled, and non-facts can be constructively generated to match surface statistics while preserving sufficient randomness. This allows us to test how pretrained priors (e.g., learned publisher-prefix structure) interact with the memorization task, in direct comparison with the structure-free random ID baseline.
>
> Preliminary results from the ISBN experiments are showing qualitatively consistent tradeoff curves, with roughly 20% KL divergence overhead relative to the corresponding theoretical optimum. Interestingly, LoRA fine-tuning exhibits a substantially lower parameter efficiency, storing only ~0.22 bits/param of information compared to the ~2 bits/param of our from-scratch models. This also supports our argument that the effective memory budget for random facts is far smaller than the raw parameter count.
>
> ---
>
> ***On finite-$p$ effects.***
>
> The reviewer is correct that our current experiments operate strictly in the n/u ≪ 1 regime. We will include in the revision experiments that vary p = n/u and compare the empirical memory cost against the first-order correction predicted by Theorem 3.3. This natural extension will test whether the theorem's characterization is empirically predictive beyond the leading-order term, and clarify how the tradeoff changes as the universe becomes less sparse.
>
> ---
>
> ***Structured knowledge and real-world applicability.***
>
> We acknowledge this as a limitation in scope. That said, structure does not eliminate the tradeoff: it reduces randomness of the facts when conditioned on background knowledge, effectively shrinking the information that must be independently memorized, which shifts the Pareto frontier rather than removing it. Crucially, factual knowledge always retains an incompressible residual (specific names, dates, identifiers) that cannot be deduced from structural regularities alone, and our theory characterizes precisely this component.
>
> What we cannot currently provide is a formal, quantitative account of how much structure compresses the effective fact count in a given domain. We will discuss this honestly in a dedicated Limitations section, and the planned ISBN experiment is also designed to probe how a pretrained prior interacts with memorization of the residual randomness.

---

> > ### Author Rebuttal · Reviewer_33hH · 2026-04-03
> >
> > Thank you for the detailed rebuttal. The proposed real-world extension using ISBN numbers and the planned experiments on finite-(p) effects address my main concerns. I also appreciate the commitment to adding intuitive explanations for each theorem.
> >
> > Overall, the rebuttal strengthens the paper and provides a clearer path toward empirical validation. I encourage the authors to include the ISBN-based results, finite-(p) experiments, and added intuition in the final version. Based on these clarifications, I have increased my score.

---

### Official Review · Reviewer_N74J · 2026-03-13

**Soundness:** 4
**Presentation:** 3
**Significance:** 4
**Originality:** 4
**Overall Recommendation:** 5
**Confidence:** 4

**Summary:**

This paper is deriving a theoretical explanation for hallucinations that people see in LLMs. They explain hallucinations from an information theory perspective, specifically using rate distortion. The problem of hallucination is abstracted as a membership testing problem. A model learns from seeing the set of facts and non-facts, and has to compress this knowledge with limited amounts of memory. The task of the model is to output a probability (capturing a confidence level) on whether a particular input is a fact or non-fact. The authors show that the distribution of outputs on facts and non-facts should be similar, under KL divergence, for memory to be low. Thus, this completes their idea that it is optimal under rate distortion for models to hallucinate, specifically hallucinate with high confidence, on non-facts. The authors show some implication of these theoretical results using specific examples of loss functions and they give experimental results.

**Compliance With Llm Reviewing Policy:**

Affirmed.

**Final Justification:**

I really like this paper and think it is a great contribution to machine learning research. I did not have too many concerns (other than presentation issues) which the authors say they will try to address. I stick to my prior assessment that this paper should be accepted.

**Key Questions For Authors:**

1. Could the authors provide some directions for future work and include these in the paper?

2. Are there other information theory related treatments of hallucinations that may be related?

**Limitations:**

- Authors appropriately described the assumptions behind their results. Similar to the above question on future work, perhaps give ideas on other directions related to the hallucination problem this result does not cover.

**Strengths And Weaknesses:**

- The definitions and theorems are stated rigorously and precisely. Most proofs are in the appendix, but results seem sound..
- The implications of the results are fascinating. It can offer a good perspective to understanding hallucinations to the machine learning community and gives a direct connection of how information theory ideas can be used to analyze learning problems.  I believe this paper will have an impact.
- The paper uses solid techniques and methods. Results are original as far as I am aware.

While I like most of the presentation of the paper (for instance, the introduction motivates the paper well, related works are given, contributions are described well, and most definitions are clear), there are a few areas I think could use improvement.

- The quantities $\mu_K$ and $\mu_N$ should be defined better. It is mostly defined in English words, and I believe these quantities are the key to all the results so they deserve a mathematical definition with details explained. For instance, are all facts and non-facts drawn with equal probability? Does that matter, and if not, give an explanation. It may also help to give some examples of $\mu_K$ and $\mu_N$ for a simple model, the same way additional examples are given for $d_K$ and $d_N$.

- In general, this paper expects the reader to understand a lot of notation and definitions. To make it more accessible to a general reader, I think the paper could define more terms exactly (like supp for support, or that $P << Q$ means absolutely continuous in parts of the appendix), or even at a more basic level, what the definition of KL divergence is.

- More intuitive explanations of the results can be provided along with the theoretical statements.

- Just because I am curious, perhaps in the appendix somewhere, the authors can give details on how their result recovers the bounds in their cited work Hurley & Waldvogel (2007). It would also be interesting to explore other connections like this one.

Overall I really liked this paper and its results. I am glad I got the chance to review it.

---

> ### Author Rebuttal · Authors · 2026-03-31
>
> We thank the reviewer for the careful reading and constructive suggestions. We address the weaknesses and questions below.
>
> ---
>
> ***Formal definitions of $\mu_K$ and $\mu_N$***
>
> We thank the reviewer for pointing out the ambiguity. In the revised version, we will add formal definitions of $\mu_K$ and $\mu_N$ in the Preliminaries: let $\mathcal{K} \subseteq \mathcal{U}$ be a uniformly random subset of size $n$, and let $I \sim \mathrm{unif}(\mathcal{K})$ be a random key; then $\mu_K(\mathcal{M})$ is defined as the distribution of the random variable $\mathsf{Query}^{\mathcal{M}}(I, \mathsf{Init}^{\mathcal{M}}(\mathcal{K}))$.
>
> We will also add a concrete example to illustrate $\mu_K$ and $\mu_N$. For instance, a standard filter with FPR $\varepsilon$ satisfies $\mu_K = \delta_1$ and $\mu_N = (1-\varepsilon)\delta_0 + \varepsilon\delta_1$.
>
> ---
>
> ***Heavy notation***
>
> Accessibility has been mentioned by multiple reviewers. We will invest further effort to improve readability, adding intuition throughout and restructuring where needed. Thank you for suggesting including formal definitions of terminologies such as $P \ll Q$, and KL divergence; we will add these in the revised version.
>
> ---
>
> ***Intuition for theoretical results***
>
> We agree that the paper would benefit from more accessible exposition, and we will add a plain-language remark after each main theorem. For instance, after Theorem 3.1 we will include a paragraph explaining how the rate-distortion result reduces space lower bounding to the problem of finding optimal $\mu_K, \mu_N$, and how this relates to the filter lower bound. We will tailor each such remark to the specific result it accompanies.
>
> ---
>
> ***Relationship to the space lower bound in Hurley & Waldvogel (2007)***
>
> Thank you for mentioning this point. We plan to include a dedicated subsection discussing  the various known space bounds, including those by Carter et al. and Pagh & Rodler, and compare them with special cases of our results. Regarding Hurley & Waldvogel specifically, their space bound can be recovered by taking our rate-distortion function $R_p(\varepsilon_K, \varepsilon_N)$ in section 3 (since they consider a fixed key-to-universe ratio), and then using a weighted sum of $\varepsilon_K$ and $\varepsilon_N$ as a single distortion measure.
>
> ---
>
> ***Key Question: future directions***
>
> One natural follow-up question is the modelling of relatively structured data. For instance, what happens if the facts are correlated, say, when there are $\sqrt{|\mathcal{U}|}$ clusters of potential facts each of size $\sqrt{|\mathcal{U}|}$, and the correctness of these statements are positively correlated within the groups? It is theoretically interesting to see what certain structures can bring to the equation.
>
> Moreover, it will be interesting to further explore the information-theoretic theories for “memorization”, both theoretically and empirically. For example, the quantity $I(\mathcal{K}, W)$ is very hard to measure empirically, and it would be an interesting direction to explore what would be a good proxy in real-world LLMs.
>
> ---
>
> ***Key Question: other information-theoretic treatments of hallucination***
>
> The cited paper “Understanding LLM behaviors via compression” by Pan et al. (2025) considers memory and error (hallucination) using Kolmogorov complexity. Moreover, the paper “Predictable compression failures” by Chlon et al. (2025) studies hallucinations through a different kind of compression, which involves the language model’s input context. Other relevant compression-based views of hallucination are also mentioned in Shi et al. (2025), Mohsin et al. (2025), and Kim (2025a,b), which we briefly mentioned in the related works section.

---

> > ### Author Rebuttal · Reviewer_N74J · 2026-04-03
> >
> > Thanks for the detailed rebuttal. Other than improving presentation (which the authors addressed), I did have any other serious concerns about the work. I maintain that it should be accepted.

---

### Decision · Program_Chairs · 2026-04-30

**Decision:**

Accept (spotlight)

**Comment:**

This paper takes an original perspective of looking at LLM hallucinations through the lens of information theory. It abstracts the problem of hallucination as a membership testing problem (fact vs. non-fact) and derives an nice theoretical insight that to optimise for memory costs, the distributions of confidence estimates which a model assigns to facts and non-facts should be similar. In the process, it provides an explanation for hallucinations. The theoretical result is validated in admittedly very restricted settings (small transformers, synthetic string), but the community could benefit from the new perspectives offered by the paper.